# Collective interactions among organometallics are exotic bonds hidden on lab shelves

Shahin Sowlati-Hashjin[1], Vojtěch Šadek[2,3], SeyedAbdolreza Sadjadi[4], Mikko Karttunen [5,6,7],
Angel Martín-Pendás [8✉] & Cina Foroutan-Nejad [9✉]

Recent discovery of an unusual bond between Na and B in $NaBH_3^-$ motivated us to look for potentially similar bonds, which remained unnoticed among systems isoelectronic with $NaBH_3^-$. Here, we report a novel family of collective interactions and a measure called exchange-correlation interaction collectivity index ($ICI_{XC}$; $ICI \in [0,1]$) to characterize the extent of collective versus pairwise bonding. Unlike conventional bonds in which $ICI_{XC}$ remains close to one, in collective interactions $ICI_{XC}$ may approach zero. We show that collective interactions are commonplace among widely used organometallics, as well as among boron and aluminum complexes with the general formula $[M^{a+}AR_3]^{b-}$ (A: C, B or Al). In these species, the metal atom interacts more efficiently with the substituents (R) on the central atoms than the central atoms (A) upon forming efficient collective interactions. Furthermore, collective interactions were also found among fluorine atoms of $XF_n$ systems (X: B or C). Some of organolithium and organomagnesium species have the lowest $ICI_{XC}$ among the more than 100 studied systems revealing the fact that collective interactions are rather a rule than an exception among organometallic species.

[1] Institute of Biomedical Engineering, University of Toronto, Toronto, Ontario M5S 3G9, Canada. [2] CEITEC—Central European Institute of Technology, Masaryk University, Kamenice 5, CZ-62500 Brno, Czechia. [3] Department of Chemistry, Faculty of Science, Masaryk University, Kamenice 5, CZ-62500 Brno, Czechia. [4] Department of Physics, Faculty of Science, Laboratory for Space Research, The University of Hong Kong, Hong Kong SAR, China. [5] Department of Chemistry, The University of Western Ontario, 1151 Richmond Street, London, Ontario N6A 3K7, Canada. [6] Department of Physics and Astronomy, The University of Western Ontario, 1151 Richmond Street, London, Ontario N6A 5B7, Canada. [7] Centre for Advanced Materials and Biomaterials Research, The University of Western Ontario, 1151 Richmond Street, London, Ontario N6K 3K7, Canada. [8] Departamento de Química Física y Analítica, University of Oviedo, 33006 Oviedo, Spain. [9] Institute of Organic Chemistry, Polish Academy of Sciences, Kasprzaka 44/52, 01-224 Warsaw, Poland. ✉email: ampendas@uniovi.es; cforoutan-nejad@icho.edu.pl

The nature of the Na–B bond in $NaBH_3^-$ has been perhaps the most controversial topic among the chemical bond community since the beginning of the COVID-19 era[1–7]. Previous studies on this bond and M–B bonds of other $[M^{n+}BH_3]^{2-n}$ complexes (M: Li, Na, K, Mg, and Ca) within the context of the most sophisticated bond analysis methodologies have revealed that these bonds are indeed unique[4–8]. In a recent work Radenkovic et al.[6] used breathing orbital valence bond (BOVB) analysis and verified that only $-3.6$ kcal.mol$^{-1}$, i.e., 9.9% of the bond dissociation energy ($-36.4$ kcal.mol$^{-1}$) in Na–B bond of $NaBH_3^-$ originates from the spin-exchange covalent bonding mechanism that corresponds to one-electron bonding[5] while the major contribution is electrostatic, as originally suggested by Foroutan-Nejad[4]. Thus, the main difference between the M–B bonds of the $[M^{n+}BH_3]^{2-n}$ species and other more conventional ones is that their metals are not bonded merely to the boron in the $BH_3$ fragment. Instead, the metal interacts strongly with the hydrogens in the $BH_3$ fragment (or the CN moiety in $NaB(CN)_3^-$)[4]. In other words, in a classical Lewis picture of these compounds, the 1,2 M–B interactions in $[M^{n+}BH_3]^{2-n}$ are either destabilizing or remarkably weakly stabilizing compared to the 1,3 M⋯H interatomic interactions, Fig. 1. In this sense, considering both 1,2 and 1,3 interactions, i.e., short- and longer-ranged contributions, is essential, as much as in the case of an ionic crystal in which we need to add the slowly converging electrostatic terms to obtain the correct Madelung constant that provides most of their lattice energies.

The above raises a fundamental question: where exactly do these bonds fit within the spectrum of the known chemical bonds? One may speculate that aluminum analogs of $NaBH_3^-$ ($[M^{n+}AlH_3]^{2-n}$), on one hand, and their carbon analogs ($[M^{n+}CR_3]^{1-n}$) on the other hand, might form similar bonds. Although the former aluminum clusters are at present a mere computational curiosity[9], the latter carbon-based group includes a large number of well-known organometallic reagents, including Grignard and organolithium reagents. Are the closest relatives of the Na–B bond in $NaBH_3^-$ already on the shelves of the chemistry labs but unnoticed by theoretical chemists?

To answer the abovementioned questions, in this work we introduce a measure called interaction collectivity index, ICI, of an atom in a molecule. ICI is a metric that characterizes the extent of an atom's pairwise interactions with its neighboring atom (vide

infra) versus the collective interactions of that atom with all the other atoms of the system. We re-examine the nature of bonding between metals and carbon atoms of some well-known organometallics and verify species with collective interactions.

## Results and discussions

The structures of species with the general formulas $[M^{n+}AH_3]^{2-n}$ (M: Li, Na, K, Mg, Ca, and Sr; A: B and Al), and $[M^{n+}AR_3]^{1-n}$ (M: Li, Na, K, Be, Mg, Ca, and Sr; R: H, $CH_3$, F, CN, and Phenyl; A: C), with boron, aluminum, and carbon as the central atoms of the complexes were optimized, and local minima with $C_{3v}$ geometrical symmetry were selected for further analysis, Fig. 1. To test the possibility of spin-polarized bonding, as suggested by Salvador et al. in the case of $NaBH_3^-$[5], all systems were also optimized by broken symmetry DFT (BS-DFT)[10]. Only seven systems among the boron and aluminum clusters were found to be more stable at the BS-DFT level. The electronic structures of these seven molecules were further analyzed at the coupled cluster (CC) and complete active space (CAS) computational levels but the general conclusions remained the same, vide infra.

Additionally, a test set including fifty-three different classical species known to form strong bonds including ionic, covalent, dative, and charge-shift bonding were also analyzed. Their bonding characteristics were compared with those of the boron and aluminum clusters and organometallics to have a comprehensive picture of bonding among a wide range of molecules. In this test set, we do not consider species with weak, noncovalent bonds. The set includes halides and oxides of alkaline and alkaline-earth metals with the general formula $M_aX_b$ (M: Li, Na, K, Be, Mg, Ca; X = O, F, and Cl), $BX_3$ (X: H, F, Cl), $X_2$ (X: H, N, O, F, Cl, and Br), $H_2X$ and $H_2X_2$ (X: O, S, Se), ethane, ethene and ethyne, $CF_4$, CO, $CN^-$, $NH_3$, $PH_3$, $N_2H_4$, $NH_4^+$, NO, $NO^+$, $NO^-$, and $NH_3BH_3$.

**Which indices can be used to safely classify bonds?** The M–B bonds in $[M^{n+}BH_3]^{2-n}$ species were found to display several seemingly unique properties within the context of quantum chemical topology approaches as previously reported by different researchers. Among them[1,4–7]:

(1) One $(3, -1)$ critical point (CP) forms between metals and boron but no $(3, -1)$ CP between the substituents on the boron atom, and the metal at the equilibrium geometry based on the quantum theory of atoms in molecules (QTAIM) analysis[11]. The electron density at the $(3, -1)$ CP between the Na and B atoms was found to be minute, inconsistent with a covalent bond[4]. Nevertheless, as discussed by Shahbazian and co-workers[12,13], the electron density at $(3, -1)$ CP is to a large extent reproducible by the corresponding promolecule density that is the sum of the electron densities of the non-bonded free atoms[14,15]. To avoid this controversy, we prefer here energetic bonding descriptors instead of those based on the electron density and its derivatives at the $(3, -1)$ CPs.

(2) Large electron delocalization associated with the stabilizing inter-atomic exchange-correlation energy component of bonding[4].

(3) A destabilizing interatomic electrostatic energy between the metal and the boron atom[4].

(4) Stabilizing interactions, both of electrostatic and exchange-correlation nature, between the metal and the substituents around the central boron atom that is the main driving force for the formation of the molecules, Fig. 1[4].

It is worth mentioning that Radenkovic et al. discovered that 68.7% of the bond dissociation energy (25 kcal.mol$^{-1}$) in the Na–

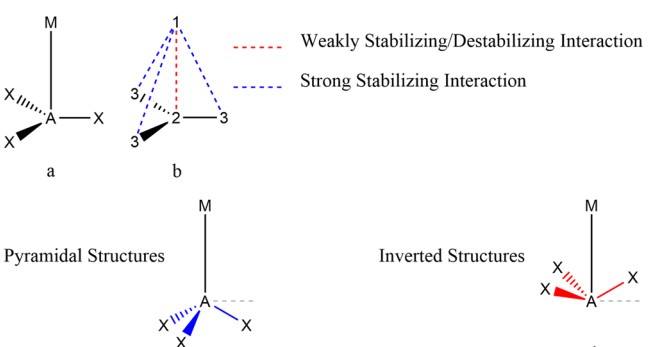

**Fig. 1 Collective bonding cannot be presented a by Lewis structure.** Schematic representations of molecular geometries (**a**) the general Lewis structure of the studied species ($[M^{a+}AX_3]^{b-}$) in this work, and (**b**) the stabilizing/destabilizing nature of interactions in $[M^{a+}AX_3]^{b-}$ clusters. While the 1,2 interactions in a Lewis structure are destabilizing or merely weakly stabilizing, the 1,3 interactions proved to be strongly stabilizing. Panels (**c**) and (**d**) provide schematic representations of the pyramidal and inverted $[M^{a+}AX_3]^{b-}$ species studied in this work. The inverted structures are marked with i- throughout the article and are distinguishable by their negative $\Delta\angle$M–A–X values defined as the difference between the $\angle$M–A–X angle and a rectangle (shown as a grey dashed line perpendicular to the M–A bond) as listed in Table 1, vide infra.

B bond of $NaBH_3^-$ originates from a dipole-dipole interaction in the Heitler-London resonance structure[6]. Interestingly, the Heitler-London contribution formally represents the covalent part of the wave function within the context of the valence bond theory[6]. Foroutan-Nejad traced back this contribution to the Na⋯H interactions within the context of QTAIM using the Interacting Quantum Atoms (IQA) energy partitioning method[16–19]. Herein, we focus on the characteristics 2, 3, and 4 from the above list to further analyze the bonds, emphasizing that orbital-based analyses have provided different, insightful, yet partial views of the whole picture, when trying to reconstruct a 3D object from several of its 2D projections. This orbital bias is avoided, or at least softened, when orbital invariant descriptors are used.

The equilibrium bond length is a function of the attractive and repulsive forces between the interacting atoms that determine the bond dissociation energy, $D_e$. Within the context of IQA, the $D_e$ between two fragments A and B in a molecule is the sum of three energy components, deformation energy ($E_{Def}$), promotion energy ($E_{Pro}$), and interaction energy ($E_{Int}$). The deformation energy is the energy needed to change the structure of fragments in their free form to the structures in the molecule. The promotion energy is the energy difference between individual fragments in the molecule and the energy of the fragment in the same geometry, i.e., the energy needed for electron reorganization. Finally, the interaction energy is the stabilizing part of the energy between two fragments in the molecule. Supplementary Table 1 lists all energy components along with bond dissociation energies. Please note that the interaction energy and its components are state functions within the context of IQA[20,21]. The magnitudes of promotion and deformation energies may change as a result of the energy difference between the selected reference state and the electronic state of the fragment in the molecule. The interatomic interaction energy is further dissected into a classical Coulombic, or electrostatic, term, $V_C(A, B)$, that can be read in chemical terms as an ionic contribution, and a quantum mechanical exchange-correlation component, that corresponds to covalency, *vide infra*. Of these energy components, $V_{XC}(A, B)$ is always negative (stabilizing) between all pairs of atoms in a molecule at the equilibrium geometry. Destabilizing electrostatic energy between two covalently bonded atoms is not uncommon but it is exclusive to homonuclear bonds or covalent bonds between atoms with close electronegativities. For instance, $V_C(A, B)$ between the nitrogen atoms in $N_2$ and C–H of ethane are +133.9 and +23.6 kcal.mol$^{-1}$, respectively (see Supplementary Table 2). In polar covalent bonds, in addition to the stabilizing effect of the exchange-correlation energy component, the Coulombic energy component is also strongly stabilizing, e.g., in $CN^-$, $V_{XC}(C, N)$ and $V_C(C, N)$ are −374.1 and −669.0 kcal.mol$^{-1}$, respectively. In ionic bonds, the contribution of $V_{XC}(A, B)$ is small and stabilizing while the electrostatic part is dominant, e.g., in KCl, $V_{XC}(K, Cl)$, and $V_C(K, Cl)$ are −31.0 and −104.7 kcal.mol$^{-1}$, respectively. It is worth re-emphasizing that $V_{XC}(A, B)$ is related to the extent of electron-sharing through orbital overlaps between any pair of atoms that is a direct measure of covalency[22–24]. To have an overview of the variation of bond characteristics among a wide range of bonds, we plotted $V_{XC}(A, B)$ and $V_C(A, B)$ values of 103 AB bonds as a two-dimensional space, Fig. 2.

Conventional covalent, ionic, and polar covalent bonds in our test set (marked with blue diamonds) can be distinguished easily in Fig. 2. Nonpolar covalent bonds are accumulated in the lower corner of the right-hand side of the plot, where the $V_C(A, B)$ component of the dissociation energy is positive. All these bonds have an exclusive stabilizing contribution from the interatomic exchange-correlation energy component. The least stabilizing

interatomic exchange-correlation energy for a non-polar covalent bond in our test set is that of the Se–Se bond in $H_2Se_2$, which is merely −94.4 kcal.mol$^{-1}$; we arbitrarily choose this value as the lowest threshold of $V_{XC}(A, B)$ for covalency. This threshold is marked by a dashed red line in Fig. 2. Consequently, the species having higher stabilizing (more negative) interatomic exchange-correlation energies fall within the realm of the covalently bonded systems. By this choice of reference, BeO, MgO, and CaO also belong to the regime of covalently bonded species although these species have substantially larger contributions from $V_C(A, B)$ ensuring the dominance of ionicity in their bonding, i.e., polar covalency (see Supplementary Table 2 in which the studied molecules are sorted based on the magnitudes of their $V_{XC}(A, B)$ and $V_C(A, B)$). It is worth emphasizing that some diatomic molecules formed by vaporization of ionic crystals indeed sustain covalent character in the gas phase[25].

Formally, ionic bonds can also be distinguished by their low $V_{XC}(A, B)$ energies and large stabilizing $V_C(A, B)$ contributions (see Fig. 2 and Supplementary Table 2). KCl/LiF with $V_{XC}(A, B)$ energies of about −31 kcal.mol$^{-1}$ are selected as the prototypes of ionic bonding. This is done because these bonds are formed between cations and anions that have similar hardness, *e.g.*, a soft-soft interaction between $K^+$ and $Cl^-$, and a hard-hard interaction between $Li^+$ and $F^-$, both lead to the same amount of $V_{XC}(A, B)$. Accordingly, any species with $V_{XC}(A, B)$ less stabilizing than −31 kcal.mol$^{-1}$ can be classified as ionic in our test set. A number of molecules still fall within the "grey-zone" where it is hard to ascribe a pure covalent or ionic nature to their bonding. Species such as KF, $BeCl_2$, $MgCl_2$, $MgF_2$, $BeF_2$, $CaCl_2$, $K_2O$, $CaF_2$, $NH_3$–$BH_3$, $BF_3$, and $BCl_3$ lie on this spectrum and all have interatomic exchange-correlation energies that fall between the thresholds of ionic (KCl/LiF) and covalent ($H_2Se_2$). The bonding spectrum in this zone starts from the polarized-ion ionic bonds and ends with the strongly polarized covalent bonds.

The so-called charge-shift bonds (CSB) introduced within the framework of valence bond theory[26] are also distinguishable in this plot by larger stabilizing $V_{XC}(A, B)$ energy components compared to ordinary covalent bonds (see Supplementary Table 1 for sorted numerical data). While an ordinary single covalent bond, such as the C–C bond in ethane, has an absolute (note that the numbers are negative) stabilizing interatomic exchange-correlation contribution of only −189.6 kcal.mol$^{-1}$, the contribution of the $V_{XC}(A, B)$ component for the charge-shift bonds ($F_2 = -227$, HO–OH = −225, $H_2N$–$NH_2 = -220$ kcal.mol$^{-1}$) is clearly more stabilizing by at least 30 kcal.mol$^{-1}$.

### $[M^{n+}BH_3]^{2-n}$, $[M^{n+}AlH_3]^{2-n}$, and some organometallics conform to neither covalency nor ionicity.

Figure 2 shows that the M–B interaction in $[M^{n+}BH_3]^{2-n}$ species fall into the top right corner of the $V_{XC}(A, B)$ vs. $V_C(A, B)$ plot. This is a region that none of the conventional bonds in our test set occupies. Among these species, $V_C(A, B)$ is strongly destabilizing akin to nonpolar covalent bonds, but $V_{XC}(A, B)$ is not as stabilizing as in conventional covalent bonds. Here, we re-emphasize that in the $[M^{n+}BH_3]^{2-n}$ systems the M–B interactions, unlike interactions in nonpolar covalent bonds that are formed between two nonmetals with similar electronegativities, are between a metal and a non-metal. All $[M^{n+}AlH_3]^{2-n}$ species except for $KAlH_3^-$ have slightly destabilizing $V_C(M, Al)$. The slight stabilization (−0.9 kcal.mol$^{-1}$) in $KAlH_3^-$ may originate from the substantial charge transfer between K and Al that is more pronounced compared to all other $[M^{n+}AlH_3]^{2-n}$ species, Table 1.

The trends in the variations of the metal charges in $[M^{n+}AlH_3]^{2-n}$ complexes correspond to their boron counterparts. However, unlike in the $[M^{n+}BH_3]^{2-n}$ complexes, the total interatomic interaction

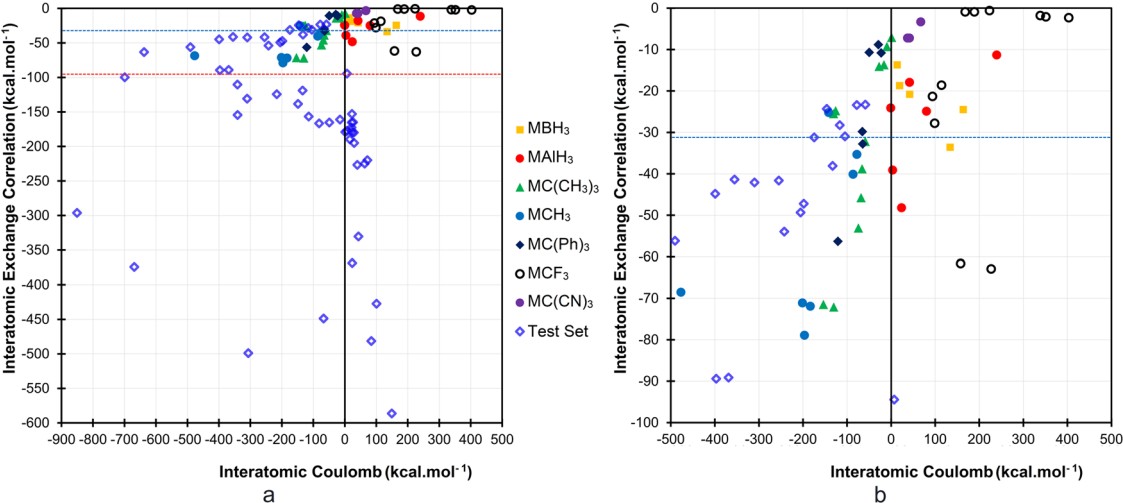

**Fig. 2 Plotting the bonding energy components versus each other reveals the nature of bonding.** Interatomic exchange-correlation energy, $V_{XC}(A, B)$, versus interatomic Coulomb energy, $V_C(A, B)$. (**a**) The full plot and (**b**) a part of the original plot focusing on organometallics. The two blue and red dashed lines mark boundaries with $V_{XC}(A, B) = -31\,\text{kcal.mol}^{-1}$ (corresponding to the interatomic exchange-correlation energy component of KCl/LiF, the upper limit of $V_{XC}(A, B)$ for ionic interactions) and $V_{XC}(A, B) = -94.4\,\text{kcal.mol}^{-1}$ that is the lower limit of $V_{XC}(A, B)$ for conventionally known covalent bonds (Se–Se bond in $H_2Se_2$), respectively. See Supplementary Table 2 for numerical data and the text for details.

energies between the Li, Na, and K atoms and aluminum are stabilizing because the exchange-correlation energy component dominates the destabilizing (slightly stabilizing for K-Al interaction) interatomic Coulombic energy. The M-Al interaction for the Mg and Ca complexes is destabilizing because of the large positive electrostatic energy component that masks $V_{XC}(M, Al)$, Table 1. The interaction between the metals and hydrogens in $[M^{n+}AlH_3]^{2-n}$ complexes is significantly stabilizing for M: Li, Mg, and Ca, and is electrostatic in nature. The M···H interaction is the sole stabilizing interaction in $[M^{n+}AlH_3]^{2-n}$ (M: Mg and Ca) complexes akin to $[M^{n+}BH_3]^{2-n}$ species as discussed elsewhere [4].

Seven species including all $[M^{n+}BH_3]^{2-n}$ clusters and two $[M^{n+}AlH_3]^{2-n}$ (M: Na and K) were identified by the T-diagnostic test at the CCSD(T)/def2-SVP[27] level to have notable multi-reference character at their equilibrium geometries. The wave functions of these molecules were thus also analyzed at the CCSD/def2-SVP, CASSCF(8, 8)/def2-SVP, and M06-2X/def2-SVP levels at the CCSD/def2SVP geometries. The magnitudes of each energy component slightly change at different levels of theory, but the general trends remain the same, Supplementary Table 3-5. Therefore, our conclusions based on DFT remain intact and unchanged. The magnitude of $V_{XC}(M, A)$ increases from CAS to CC and then to DFT (performed at CCSD/def2-SVP optimized geometry) in line with the increase in the dynamic correlation and HF exchange at the M06-2X DFT level. Interestingly, the $V_{XC}(M, A)$ values of the optimized DFT structures are reasonably close to those obtained at the CCSD and CAS levels of theory, Supplementary Tables 3 and 4. This confirms that the M06-2X/ def2-TZVPP geometry and wave function are a safe road towards bonding analysis among these species using the IQA approach.

The M-C interactions in M-CF_3 and $i$-M-C(CN)_3 fall next to the M-B and M-Al interactions of boron and aluminum clusters in the same region of Fig. 2. The M-C interactions in $i$-MC(CN)_3 and the high-energy inverted isomers of MCF_3, $i$-MCF_3, (see Fig. 1 for the definition of inverted versus pyramidal) have extremely destabilizing $V_C(M, C)$ values and relatively negligible $V_{XC}(M, C)$ contributions. The Mg-C interaction in the high-energy local minimum with inverted CF_3 ($i$-MgCF_3) has the largest destabilizing $V_C(M, A)$ (403.1 kcal.mol$^{-1}$) of all studied systems with negligible $V_{XC}(Mg, C)$ ($-2.3$ kcal.mol$^{-1}$). On the

other hand, the lowest energy isomer of MgCF_3 (12.4 kcal.mol$^{-1}$ lower in energy than $i$-MgCF_3) has large $V_{XC}(Mg, C)$ ($-63$ kcal.mol$^{-1}$) comparable to that of the B-F interaction in BF_3, which is conventionally thought of as a polar covalent bond[28,29]. The interatomic exchange-correlation energy components between the metals and the central carbon in MCF_3 or $i$-MC(CN)_3 are not large enough to compensate for the strong destabilizing electrostatic energy between these atoms as it is reflected in the destabilizing nature of $V_{Int}(M, A)$, Table 1. Once again, the origin of bonding in these molecules is the strong interaction between the metal ions and the substituents (F or CN) found around the central atoms as has been similarly discussed for $[M^{n+}BH_3]^{2-n}$ species[4]. The M-C interactions in several MC(CH_3)_3 and MC(Ph)_3 molecules have negative but close to zero $V_C(M, C)$ values comparable to those of MCF_3, MC(CN)_3, as well as their boron and aluminum analogs, Fig. 2 and Table 1.

**Pairwise versus collective interactions.** Thus far, we have shown that the pairwise interactions between M-B, M-Al, or M-C in the MCF_3 and MC(CN)_3 species are either weakly stabilizing or even completely destabilizing. These molecules are thus formed because of the stabilizing interactions between the metal and the hydrogens or other substituents on the periphery of the central atoms in BH_3, AlH_3, CF_3, or C(CN)_3. In this sense, the metal atoms form "collective" interactions in $[M^{n+}BH_3]^{2-n}$, $[M^{n+}AlH_3]^{2-n}$, and $[M^{n+}CX_3]^{1-n}$ (X = F or CN), whereas all other bonds in these systems are strong (covalent and/or ionic) interactions that involve exchange-correlation contributions between just the given pair of atoms [24].

To assess the nature of the M-A interactions in $[M^{a+}AX_3]^{b-}$ complexes, we define the exchange-correlation interaction collectivity index for atom Y, denoted $ICI_{XC}(Y)$, as the ratio between $V_{XC}(Y,\{M\})$, where $\{M\}$ is the set of all 1,2 neighboring atoms, and $V_{XC}(Y,\{T\})$, where $\{T\}$ stands for the set of all atoms of the system except Y, $ICI_{XC}(Y) = \frac{V_{XC}(Y,\{M\})}{V_{XC}(Y,\{T\})}$. Note that the term "neighbor" refers to 1,2 neighboring atoms in the simple formal Lewis structure presented in Fig. 1. When an atom has solely a single neighbor (Z), and when there is a bond between them, discerned or assumed by whatever means, then the above ratio

**Table 1 The $\Delta\angle$M–A–X angle, QTAIM atomic charges of M and A in MAX$_3$ systems, Q(M) and Q(A), the IQA interaction energy and its interatomic exchange-correlation and Coulombic energy components between metal and Al, B, or central C atoms in the studied systems, $V_{Int}$(M,A), $V_{XC}$(M, A) and $V_C$(M, A), and those of the metals with the substituents on the central atom, $V_{XC}$(M, X) and $V_C$(M, X), in kcal.mol$^{-1}$.**

| Molecules | $\Delta\angle$M–A–X | Q(M) | Q(A) | Q(X) | $V_{XC}$(M,A) | $V_{XC}$(M,X) | $V_C$(M,A) | $V_C$(M,X) | $V_{Int}$(M,A) | $V_{Int}$(M,X) |
|---|---|---|---|---|---|---|---|---|---|---|
| LiBH$_3^{-a}$ | 5.11 | −0.010 | 1.326 | −0.772 | −20.8 | −4.9 | 42.1 | −27.8 | 21.3 | −32.7 |
| NaBH$_3^{-a}$ | 4.59 | −0.195 | 1.524 | −0.776 | −18.7 | −6.6 | 19.3 | −9.5 | 0.6 | −16.1 |
| KBH$_3^{-a}$ | 3.11 | −0.238 | 1.585 | −0.781 | −13.7 | −6.0 | 13.8 | −6.3 | 0.1 | −12.3 |
| MgBH$_3$ | 3.36 | 0.468 | 1.627 | −0.698 | −33.6 | −11.9 | 133.9 | −58.2 | 100.3 | −70.1 |
| CaBH$_3^a$ | 0.91 | 0.613 | 1.572 | −0.729 | −24.5 | −11.1 | 163.7 | −76.9 | 139.2 | −88.0 |
| LiAlH$_3^-$ | 16.27 | 0.206 | 1.241 | −0.816 | −48.1 | −2.8 | 23.6 | −27.0 | −24.5 | −29.8 |
| NaAlH$_3^{-a}$ | 14.86 | −0.135 | 1.587 | −0.817 | −39.1 | −3.4 | 3.5 | −2.1 | −35.6 | −5.5 |
| KAlH$_3^{-a}$ | 13.13 | −0.204 | 1.681 | −0.825 | −24.1 | −2.7 | −0.9 | 0.4 | −25.0 | −2.3 |
| MgAlH$_3$ | 2.83 | 0.172 | 2.112 | −0.761 | −17.9 | −7.3 | 41.9 | −13.7 | 23.9 | −21.0 |
| CaAlH$_3$ | 3.73 | 0.352 | 1.986 | −0.779 | −24.9 | −6.5 | 80.2 | −30.2 | 55.3 | −36.7 |
| $i$–CaAlH$_3$$^{a,b,d}$ | −33.89 | 0.887 | 1.471 | −0.786 | −11.3 | −24.1 | 239.5 | −128.9 | 228.3 | −153.0 |
| LiCF$_3$ | 25.14 | 0.918 | 1.197 | −0.705 | −21.3 | −0.7 | 93.8 | −78.9 | 72.5 | −79.6 |
| NaCF$_3$ | 25.29 | 0.865 | 1.261 | −0.709 | −27.8 | −1.1 | 98.8 | −67.7 | 71.0 | −68.8 |
| BeCF$_3$ | 19.12 | 1.625 | 1.303 | −0.642 | −61.6 | −3.4 | 157.8 | −151.3 | 96.2 | −154.7 |
| MgCF$_3$$^{+c}$ | 19.78 | 1.453 | 1.512 | −0.655 | −62.9 | −4.5 | 227.0 | −120.4 | 164.1 | −124.9 |
| $i$–LiCF$_3$$^{b,c,d}$ | −31.33 | 0.932 | 1.203 | −0.712 | −0.6 | −9.1 | 223.2 | −122.8 | 222.6 | −132.0 |
| $i$–NaCF$_3$$^{b,c,d}$ | −30.18 | 0.934 | 1.220 | −0.718 | −0.9 | −10.1 | 188.8 | −103.8 | 187.9 | −113.9 |
| $i$–KCF$_3$$^{b,d}$ | −29.67$^c$ | 0.942 | 1.215 | −0.719 | −0.9 | −11.6 | 168.0 | −93.1 | 167.1 | −104.7 |
| $i$–MgCF$_3$$^{+b,d}$ | −33.31 | 1.830 | 1.229 | −0.686 | −2.3 | −19.2 | 403.1 | −234.2 | 400.8 | −253.4 |
| $i$–CaCF$_3$$^{+b,d}$ | −32.12 | 1.812 | 1.219 | −0.677 | −2.1 | −24.5 | 351.1 | −194.7 | 349.0 | −219.1 |
| $i$–SrCF$_3$$^{+b,d}$ | −31.72 | 1.838 | 1.221 | −0.686 | −1.8 | −23.6 | 338.3 | −188.6 | 336.5 | −212.2 |
| $i$–MgC(CN)$_3$$^{+b}$ | −26.32 | 1.776 | 0.243 | −0.340 | −3.3 | −19.5 | 67.1 | −122.5 | 63.8 | −142.0 |
| $i$–CaC(CN)$_3$$^{+b}$ | −19.53 | 1.727 | 0.184 | −0.304 | −7.2 | −23.5 | 41.5 | −80.0 | 34.3 | −103.5 |
| $i$–SrC(CN)$_3$$^{+b}$ | −17.83 | 1.756 | 0.172 | −0.309 | −7.2 | −22.3 | 37.1 | −75.1 | 29.9 | −97.4 |
| LiCH$_3$ | 22.64 | 0.899 | −0.609 | 0.097 | −25.2 | −0.5 | −141.8 | −7.9 | −167.0 | −8.4 |
| NaCH$_3$ | 20.98 | 0.757 | −0.536 | −0.074 | −40.1 | −1.1 | −86.3 | −4.3 | −126.4 | −5.4 |
| KCH$_3$ | 22.09 | 0.805 | −0.516 | −0.096 | −35.3 | −1.0 | −77.8 | −6.2 | −113.1 | −7.2 |
| BeCH$_3^+$ | 19.08 | 1.669 | −0.971 | 0.101 | −68.5 | −1.5 | −476.9 | 30.8 | −545.4 | 29.3 |
| MgCH$_3^+$ | 16.63 | 1.397 | −0.605 | 0.069 | −78.9 | −1.9 | −196.6 | 16.4 | −275.5 | 14.5 |
| CaCH$_3^+$ | 22.07 | 1.618 | −0.621 | 0.001 | −71.1 | −1.9 | −201.3 | 4.0 | −272.4 | 2.1 |
| SrCH$_3^+$ | 22.03 | 1.616 | −0.587 | −0.010 | −71.9 | −1.9 | −183.5 | 1.9 | −255.4 | 0.0 |
| LiC(Ph)$_3$ | −6.73 | 0.894 | −0.218 | −0.225 | −10.7 | −5.5 | −49.6 | −31.0 | −60.3 | −36.5 |
| NaC(Ph)$_3$ | −5.35 | 0.907 | −0.161 | −0.249 | −8.8 | −6.8 | −29.0 | −29.4 | −37.8 | −36.2 |
| KC(Ph)$_3$ | −3.84 | 0.894 | −0.142 | −0.250 | −10.8 | −10.1 | −22.7 | −25.2 | −33.5 | −35.3 |
| MgC(Ph)$_3^+$ | 10.67 | 1.088 | −0.391 | 0.101 | −56.3 | −10.0 | −120.9 | 7.6 | −177.2 | −2.4 |
| CaC(Ph)$_3^+$ | −6.42 | 1.554 | −0.214 | −0.113 | −29.8 | −26.0 | −65.4 | −47.4 | −95.2 | −73.4 |
| SrC(Ph)$_3^+$ | −4.37 | 1.561 | −0.214 | −0.116 | −32.8 | −25.8 | −64.5 | −41.9 | −97.3 | −67.7 |
| LiC(CH$_3$)$_3^c$ | 21.40 | 0.882 | −0.470 | −0.137 | −24.8 | −0.9 | −125.9 | −9.2 | −150.7 | −10.1 |
| NaC(CH$_3$)$_3^c$ | 20.60 | 0.716 | −0.329 | −0.129 | −38.8 | −2.0 | −65.8 | −6.7 | −104.6 | −8.7 |
| KC(CH$_3$)$_3^c$ | 21.18 | 0.799 | −0.295 | −0.168 | −32.2 | −1.6 | −58.8 | −10.2 | −91.0 | −11.8 |
| BeC(CH$_3$)$_3^+$ | 16.88 | 1.417 | −0.772 | 0.119 | −79.2 | −3.8 | −357.6 | 27.2 | −436.8 | 23.4 |
| MgC(CH$_3$)$_3^+$ | 15.52 | 1.136 | −0.428 | 0.097 | −72.1 | −5.0 | −130.2 | 15.2 | −202.3 | 10.2 |
| CaC(CH$_3$)$_3^+$ | 19.04 | 1.539 | −0.412 | −0.043 | −75.5 | −3.7 | −153.6 | −5.6 | −225.3 | −9.3 |
| $i$–LiC(CH$_3$)$_3$$^d$ | −12.92 | 0.799 | −0.197 | −0.201 | −14.1 | −11.6 | −27.1 | −32.3 | −41.2 | −43.9 |
| $i$–NaC(CH$_3$)$_3$$^{b,d}$ | −11.21 | 0.754 | −0.125 | −0.210 | −9.3 | −12.8 | −9.6 | −26.1 | −18.9 | −38.9 |
| $i$–KC(CH$_3$)$_3$$^{b,d}$ | −7.01 | 0.783 | −0.162 | −0.207 | −13.7 | −14.2 | −16.4 | −21.7 | −30.1 | −36.0 |
| $i$–BeC(CH$_3$)$_3$$^{+c,d}$ | −16.98 | 1.520 | −0.274 | −0.082 | −25.5 | −31.0 | −130.6 | −81.3 | −156.1 | −112.3 |
| $i$–MgC(CH$_3$)$_3$$^{+b,d}$ | −5.76 | 0.300 | 0.033 | 0.222 | −7.1 | −10.9 | 1.1 | 4.9 | −6.1 | −6.0 |
| $i$–CaC(CH$_3$)$_3$$^{+c,d}$ | −3.55 | 1.460 | −0.230 | −0.077 | −45.8 | −24.9 | −68.2 | −25.1 | −114 | −50.0 |
| $i$–SrC(CH$_3$)$_3$$^{+e,d}$ | −0.32 | 1.466 | −0.247 | −0.073 | −53.1 | −23.0 | −74.3 | −20.2 | −127.4 | −43.2 |

The prefix $i$– denotes structures with an inverted geometry that have a negative M–A–X angle. The $\Delta\angle$M–A–X angle is defined as the difference between $\angle$M–A–X angle and a right angle as defined in ref. [9].
$^a$Data obtained from BS-DFT calculations.
$^b$The metal forms multiple (3,-1) CPs with the AX$_3$ fragment.
$^c$The global minimum of the molecule.
$_d$- represents inverted structures, see Fig. 1.
$^e$The global minimum of the molecule has a C$_s$ point group, 1.6 kcal.mol$^{-1}$ lower in energy; therefore, it is not discussed here.

**Table 2 The exchange-correlation and Coulomb energy components in kcal.mol$^{-1}$ between selected atoms, Y, (in parentheses) and all other atoms in the molecule (T).**

| Molecules | $V_{XC}(Y,T)$ | $ICI_{XC}$ | $V_C(Y,T)$ | $ICI_C$ | Molecules | $V_{XC}(Y,T)$ | $ICI_{XC}$ | $V_C(Y,T)$ | $ICI_C$ |
|---|---|---|---|---|---|---|---|---|---|
| (Li) $LiBH_3^-$ | −35.3 | 0.589 | −41.5 | −1.014 | (K) $i\text{-}KC(CH_3)_3^a$ | −56.3 | 0.243 | −81.5 | 0.201 |
| (Na) $NaBH_3^-$ | −38.3 | 0.488 | −9.2 | −2.098 | (Be) $i\text{-}BeC(CH_3)_3^+$ | −118.5 | 0.215 | −374.5 | 0.349 |
| (K) $KBH_3^-$ | −31.8 | 0.431 | −5.2 | −2.654 | (Mg) $i\text{-}MgC(CH_3)_3^{+a}$ | −39.8 | 0.178 | −15.8 | 0.070 |
| (Mg) $MgBH_3$ | −69.3 | 0.485 | −40.7 | −3.290 | (Ca) $i\text{-}CaC(CH_3)_3^+$ | −120.5 | 0.380 | −143.5 | 0.475 |
| (Ca) $CaBH_3$ | −57.8 | 0.424 | −67.2 | −2.436 | (Sr) $i\text{-}SrC(CH_3)_3^+$ | −122.1 | 0.435 | −134.9 | 0.551 |
| (Li) $LiAlH_3^-$ | −56.6 | 0.850 | −57.4 | −0.411 | (Li) $LiC(Ph)_3$ | −27.2 | 0.393 | −142.9 | 0.347 |
| (Na) $NaAlH_3^-$ | −49.2 | 0.795 | −2.8 | −1.250 | (Na) $NaC(Ph)_3$ | −29.4 | 0.299 | −117.4 | 0.247 |
| (K) $KAlH_3^-$ | −32.2 | 0.748 | 0.1 | −9.000 | (K) $KC(Ph)_3$ | −41.1 | 0.263 | −98.5 | 0.230 |
| (Mg) $MgAlH_3$ | −39.7 | 0.451 | 0.7 | 59.857 | (Mg) $MgC(Ph)_3^+$ | −86.4 | 0.652 | −98.2 | 1.231 |
| (Ca) $CaAlH_3$ | −44.4 | 0.561 | −10.4 | −7.712 | (Ca) $CaC(Ph)_3^+$ | −108.0 | 0.276 | −208.3 | 0.314 |
| (Ca) $i\text{-}CaAlH_3^a$ | −83.6 | 0.135 | −147.3 | −1.626 | (Sr) $SrC(Ph)_3^+$ | −110.1 | 0.298 | −190.0 | 0.339 |
| (Li) $LiCF_3$ | −23.4 | 0.910 | −142.8 | −0.657 | (Cl) $BCl_3$ | −138.2 | 0.647 | −236.8 | 1.676 |
| (Na) $NaCF_3$ | −31.2 | 0.891 | −104.5 | −0.945 | (Cl) $BeCl_2$ | −52.6 | 0.786 | −251.7 | 1.412 |
| (Be) $BeCF_3^+$ | −71.7 | 0.859 | −296.3 | −0.533 | (F) $BeF_2$ | −53.2 | 0.842 | −287.0 | 1.391 |
| (Mg) $MgCF_3^+$ | −76.3 | 0.824 | −134.3 | −1.690 | (F) $BF_3$ | −112.1 | 0.565 | −376.7 | 1.693 |
| (Li) $i\text{-}LiCF_3^a$ | −27.9 | 0.022 | −145.2 | −1.537 | (H) $C_2H_2$ | −185.5 | 0.970 | 32.8 | 0.882 |
| (Na) $i\text{-}NaCF_3^a$ | −31.2 | 0.029 | −122.6 | −1.540 | (H) $C_2H_4$ | −189.5 | 0.949 | 26.9 | 0.888 |
| (K) $i\text{-}KCF_3^a$ | −35.7 | 0.025 | −111.3 | −1.509 | (H) $C_2H_6$ | −189.8 | 0.933 | 23.4 | 0.889 |
| (Mg) $i\text{-}MgCF_3^{+a}$ | −59.9 | 0.038 | −299.5 | −1.346 | (F) $CF_4$ | −187.4 | 0.649 | −287.0 | 1.863 |
| (Ca) $i\text{-}CaCF_3^{+a}$ | −75.6 | 0.028 | −233.0 | −1.507 | (Cl) $CaCl_2$ | −47.8 | 0.988 | −145.9 | 1.359 |
| (Sr) $i\text{-}SrCF_3^{+a}$ | −72.6 | 0.025 | −227.5 | −1.487 | (F) $CaF_2$ | −54.5 | 0.990 | −180.5 | 1.343 |
| (Mg) $MgC(CN)_3^{+a}$ | −61.1 | 0.054 | −300.5 | −0.223 | (H) $H_2O$ | −124.9 | 0.995 | −126.7 | 1.706 |
| (Ca) $CaC(CN)_3^{+a}$ | −80.4 | 0.090 | −198.6 | −0.209 | (H) $H_2Se$ | −158.6 | 0.964 | 27.2 | 0.833 |
| (Sr) $SrC(CN)_3^{+a}$ | −77.6 | 0.093 | −188.7 | −0.197 | (H) $H_2S$ | −183.8 | 0.975 | 2.5 | 0.556 |
| (Li) $LiCH_3$ | −26.8 | 0.940 | −165.5 | 0.857 | (H) $H_2O_2$ | −121.1 | 0.983 | −126.6 | 1.059 |
| (Na) $NaCH_3$ | −43.3 | 0.926 | −99.3 | 0.869 | (H) $H_2S_2$ | −182.8 | 0.970 | 6.5 | 1.064 |
| (K) $KCH_3$ | −38.1 | 0.927 | −96.7 | 0.805 | (H) $H_2Se_2$ | −163.8 | 0.983 | −15.4 | 0.984 |
| (Be) $BeCH_3^+$ | −72.9 | 0.940 | −384.7 | 1.240 | (K) $K_2O$ | −49.9 | 0.989 | −148.1 | 1.387 |
| (Mg) $MgCH_3^+$ | −84.6 | 0.933 | −147.7 | 1.331 | (Li) $Li_2O$ | −36.8 | 0.995 | −205.4 | 0.379 |
| (Ca) $CaCH_3^+$ | −76.6 | 0.928 | −189.5 | 1.062 | (Cl) $MgCl_2$ | −43.7 | 0.953 | −185.0 | 1.378 |
| (Sr) $SrCH_3^+$ | −77.7 | 0.925 | −177.9 | 1.031 | (F) $MgF_2$ | −43.0 | 0.979 | −228.0 | 1.360 |
| (Li) $LiC(CH_3)_3$ | −27.4 | 0.906 | −153.5 | 0.820 | (Na) $Na_2O$ | −42.6 | 0.988 | −158.7 | 0.368 |
| (Na) $NaC(CH_3)_3$ | −44.6 | 0.870 | −86.0 | 0.765 | (H) $NH_2NH_2$ | −170.4 | 0.970 | −10.1 | 4.864 |
| (K) $KC(CH_3)_3$ | −37.0 | 0.870 | −89.6 | 0.656 | (H) $NH_3$ | −169.4 | 0.983 | −14.8 | 5.524 |
| (Mg) $MgC(CH_3)_3^+$ | −87.2 | 0.827 | −84.8 | 1.535 | (H–B) $NH_3BH_3$ | −136.9 | 0.651 | −198.4 | 1.859 |
| (Ca) $CaC(CH_3)_3^+$ | −82.6 | 0.865 | −170.3 | 0.901 | (H–N) $NH_3BH_3$ | −160.1 | 0.978 | −12.7 | 9.041 |
| (Li) $i\text{-}LiC(CH_3)_3$ | −48.9 | 0.288 | −124.0 | 0.219 | (H) $NH_4^+$ | −140.3 | 0.987 | 32.8 | −4.539 |
| (Na) $i\text{-}NaC(CH_3)_3^a$ | −47.7 | 0.195 | −87.9 | 0.109 | (H) $PH_3$ | −158.5 | 0.826 | −200.1 | 1.552 |

The bond collectivity indices for the exchange-correlation and Coulombic energy components are also listed.
$^a$M has multiple bond paths connecting that with several atoms.

describes the extent of pairwise bonding between Y and Z versus the collective global interaction between Y and the rest of the atoms.

In diatomic species, $ICI_{XC}$ is equal to 1 (i.e., the atoms are connected to each other through a pairwise bond). However, as the number of atoms in a molecule increases, $ICI_{XC}$ decreases. In general, this results from additional stabilizing exchange-correlation interaction energies between atom Y and the rest of the atoms, i.e., additional 1,3 or, in general, 1,n ($n > 2$) interactions between Y and its remaining neighbors. Nevertheless, one expects $ICI_{XC}$ to remain close to one in conventional 2-electron 2-center covalent bonds and even in ionic bonds. This is confirmed by the $ICI_{XC}$ values listed for the studied conventional bonds in which the $ICI_{XC}$ values remain larger than 0.9, Table 2. In a few species such as $BeX_2$, $BX_3$, $CF_4$, $PH_3$, or the H–B bond in $NH_3BH_3$, the $ICI_{XC}$ values deviate from 1 but remain rather large. This deviation is a result of the exchange-correlation interactions between X···X atoms through space. This phenomenon is mostly notable for the negatively charged, close-packed atoms of fluorine in $BF_3$ or $CF_4$. In these species, large F–F interatomic exchange-correlation energies compensate their strongly destabilizing Coulombic interactions, reducing $ICI_{XC}$ significantly. In fact, $ICI_{XC}$ is a tool to identify such interactions. Large exchange-correlation interaction between the 1,3 fluorine atoms in $XF_n$ molecules is reminiscent of large exchange-correlation between 1,2 atoms with charge-shift bonding as discussed above and listed in Supplementary Table 2. The $XF_n$ and other systems like $XO_n$ might constitute a new class of 1,3 interactions with CSB but this should be studied within the context of valence bond theory that is beyond the scope of the current investigation. It is worth noting that $ICI_{XC}$ remains larger than 0.9 even in bonds with a significant ionic character such as K–O in $K_2O$.

On the other hand, the $ICI_{XC}$ defined for the metal atoms in $[M^{n+}BH_3]^{2-n}$, $[M^{n+}AlH_3]^{2-n}$, and for some of the studied organometallics, significantly deviates from 1. The smallest $ICI_{XC}$ values appear in inverted structures in which the metals form more than one (3, −1) CPs with the $AX_3$ fragment. Analysis of the morphology of the Kohn-Sham molecular orbitals (MOs) shows that these species can be divided into three categories: 1) Species like $CaC(CN)_3^+$ or $SrC(CN)_3^+$ that form rather standard 2 electron-2 center (2e-2c) bonds. 2) Species such as $MC(CH_3)_3$

and $i\text{-BeCF}_3^+$ whose MOs display a clear $2e$-multicenter character. 3) The rest of systems, which do not show any traditionally bonding MO between the metal and the $AX_3$ fragment, akin to the ordinary behavior of ionic systems, Supplementary Fig. 1.

Among the pyramidal structures, the naïve MO analysis suggests that all species have formally $2e\text{-}2c$ bonds. The smallest $ICI_{XC}$ values are found in the tri-phenylmethyl organometallics. This suggests a strong through-space interaction between the metals and the phenyl groups that we understand as a sign of collective bonding. Such interactions originate from penetration of a substantial part of the HOMO into the atomic basin of the X atoms of the $MAX_3$ systems, Supplementary Fig. 2.

Since canonical MOs do not always provide clean Lewis pictures, we decided to examine to what extent the $NaBH_3^-$ system deviate from the $2e\text{-}2c$ image by performing an Adaptive Natural Density Partitioning (AdNDP) analysis in real space[30]. All the AdNDP orbitals are $2e\text{-}2c$ although the Na-B bond, in agreement with the canonical MO insights, invades the H atomic basins, Supplementary Fig. 3. The $ICI_{XC}$ values of metals in the $[M^{n+}BH_3]^{2-n}$ and $[M^{n+}AlH_3]^{2-n}$ species are notably smaller than 1, but clearly larger than those of the abovementioned organometallics. Therefore, $ICI_{XC}$ suggests that organometallic bonds closely resemble those in the $[M^{n+}BH_3]^{2-n}$ and $[M^{n+}AlH_3]^{2-n}$ species irrespective of the nature of their substituents. The pyramidal $MCF_3$ systems whose M–C bonds are characterized by small negative $V_{XC}(M, C)$ and large positive $V_C(M, C)$ values have $ICI_{XC}$ close to 1. Instead, the total Coulombic interaction between their $M^+$ and $CF_3^-$ fragments suggest that the driving force behind the formation of these species is the strong electrostatic interaction between the metals and the highly charged fluorine substituents. The inverted trifluoromethyl organometallics have the lowest $ICI_{XC}$ values among all the studied species. The QTAIM molecular graphs of these species show $(3, -1)$ CPs between the metals and the fluorine atoms, and the morphologies of the molecular orbitals in $i\text{-}MCF3$ species show that these systems have $2e$-multicenter bonds, Supplementary Fig. 1.

We would like to point out that casting our results in the orbital mould leads to a loss of information. Thus, by reading IQA or real space data through AdNDP orbitals (or HOMOs), we simply show how our results are compatible with previous insights, nothing else. The bonds analyzed here are neither two-center nor multicenter that would imply short-sighted delocalization, but, instead, they are covalently long-ranged, and the collective effect of the environment is needed to rationalize them.

Similar to the case of $ICI_{XC}(Y)$, we also define the electrostatic interaction energy collectivity index for an atom, $ICI_C(Y)$, as the equivalent ratio between $V_C(Y, \{M\})$ and $V_C(Y, \{T\})$. The $ICI_C$ values, unlike the $ICI_{XC}$ ones, can take any positive or negative real value because of the nondirectional nature of the Coulombic force. Negative $ICI_C$ values are a sign of a remarkable change in the nature of the $V_C(Y, \{M\})$ versus the $V_C(Y, \{T\})$ values. For instance, all the $ICI_C$ values among the $[M^{n+}BH_3]^{2-n}$, $[M^{n+}AlH_3]^{2-n}$ (except $MgAlH_3$ that has slightly repulsive Mg…H interactions), $i\text{-}MC(CN)_3$, $MCF_3$, and $i\text{-}MCF_3$ species are negative. Similarly, hydrogen in $NH_4^+$ has a negative $ICI_C$ value. This suggests that the H⋯H electrostatic interactions in this moiety are strongly destabilizing. Negative $ICI_C$ values thus stem from destabilizing Coulomb interactions between the metals and the central atoms (A: B, Al, or C), but strongly stabilizing Coulomb interaction between the metals and the substituents on the central atoms as shown in Table 1. A positive but larger than one $ICI_C$ is still a sign of a repulsion between 1,n ($n > 2$) neighbors. For example, in $BF_3$, the B–F bond has a strongly stabilizing Coulomb component, but the interaction between the negatively charged F atoms in $BF_3$, which are in proximity of each other, is slightly destabilizing. As a

result, $ICI_C(F) = 1.693$ in this molecule. This observation once again confirms the central role of the interactions of the metal-peripheral atoms in the binding of these species.

It is worth noting that the collective interaction does not necessarily impose inverted over pyramidal geometries, e.g., among the $[M^{n+}BH_3]^{2-n}$ species the pyramidal structure is the most stable isomer. The collective interaction is merely a measure of the extent of long-range covalent-type interaction defined on the basis of the interatomic exchange-correlation energy. However, the molecular geometry is the result of a balance between the potential energy components, i.e., the atomic and interatomic exchange-correlation a well as electrostatics interactions, and the kinetic energy of the molecule.

**To what extent the ICI values are sensitive to the nature of the used atomic basins?** Up to this point, all computations have been performed within the framework of QTAIM topological atoms, which are characterized by sharp boundaries[31]. A legitimate question arises regarding whether the proposed collective interaction and concomitant ICI values are sensitive to these specific atomic boundaries and disappear upon employing other definitions for atoms in molecules. To examine this question, we performed IQA analyses employing fuzzy atoms. Supplementary Table 6 lists the computed atomic and interatomic IQA descriptors of the molecules. Despite notable variations of the atomic charges using fuzzy atom partitions, the magnitudes of the $ICI_{XC}$ values remain within the same range of those computed using QTAIM atoms. Particularly, the metals in inverted species have the smallest $ICI_{XC}$ values. Among the pyramidal species, the triphenylmethyl organometallics, $[M^{n+}BH_3]^{2-n}$ and some $[M^{n+}AlH_3]^{2-n}$ species have the lowest $ICI_{XC}$ values, respectively. This reveals that the definition of collective interactions chosen in this work is not sensitive to the detailed nature of the specific atoms in molecules used in the IQA analysis, thus proving their novel genuine character.

It is a basic assumption of conventional chemical wisdom that all 1,2 interactions in a given Lewis structure are stabilizing. Contrarily, 1,n interactions where $n \neq 2$ are known to be either stabilizing or destabilizing[24]. The class of collective interactions occupy a niche in the hierarchy of chemical bonding in which 1,2 interactions are destabilizing because of the repulsive electrostatic interaction between both atoms. Figure 3 represents a summary of the nature of the interatomic interactions in a molecule.

The source of stabilization in collective interactions depends on the nature of the substituents and the partial charges of the metal atoms. In the case of metal dications, electrostatic interactions favor cohesion while monocations benefit from the non-negligible amount of the exchange-correlation interaction between the metal and the $AX_3$ fragment. To quantify the extent of collective versus pairwise bonding, we introduced a new measure termed exchange-correlation interaction collectivity index, $ICI_{XC}$. We showed that $ICI_{XC}$ remains close to 1 for ordinary covalent and ionic bonds, e.g., H–C in hydrocarbons or M–O in $M_2O$ metal oxides. On the contrary, $ICI_{XC}(M)$ in $MAX_3$ species, where A = B, Al, and C and M = Li, Na, K, Be, Mg, or Ca with a wide variety of X substituents, deviates considerably from 1, rather dramatically in the case of certain organometallic compounds.

Furthermore, we found that pairwise interaction energies between the M and B atoms in $[M^{n+}BH_3]^{2-n}$ (M: Li, Na, K, Mg, and Ca), the M and Al atoms in $[M^{n+}AlH_3]^{2-n}$ (M: Mg and Ca), and the M and C atoms in $MC(CN)_3$ (M: Mg, Ca, and Sr), $i\text{-}MCF_3$, $MCF_3$, $i\text{-}MC(CH_3)_3$ and $MCPh_3$ (M: Li, Na, K, Be, and Mg) are destabilizing or slightly stabilizing. In the case of destabilizing interactions, the source of destabilization is

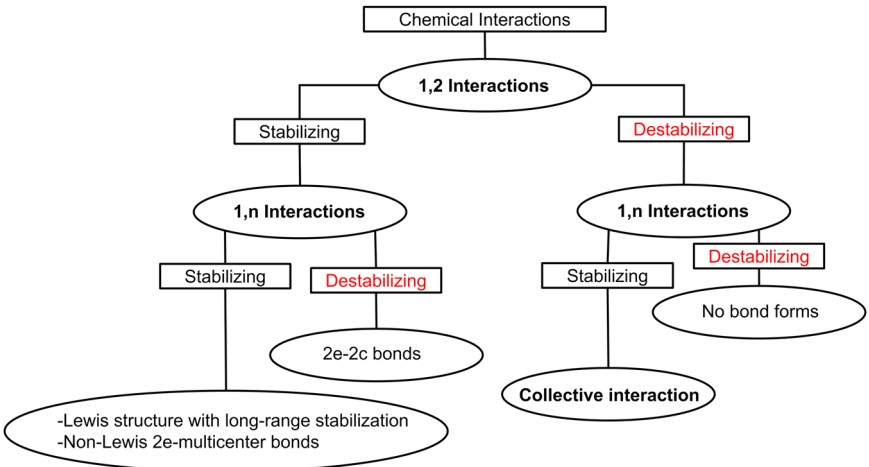

**Fig. 3 The hierarchy of chemical bonds.** The nature of the 1,2 and 1,n interatomic interactions.

electrostatics akin to nonpolar (pure) covalent bonds, while the exchange-correlation energy component is not stabilizing enough to compensate for the repulsive electrostatic component. The driving force for the formation of these $MAX_3$ complexes is then the interaction between the metals and the peripheral X substituents with a significant collective character. The relative contribution of the exchange-correlation energy and the Coulomb electrostatic interaction energy between the metals and the $AX_3$ species change from one system to the other even within the same family of molecules, e.g., $[M^{n+}BH_3]^{2-n}$. In that sense, perhaps a new name, collective bonding, can be given to these bonds which may have a dominant electrostatic or covalent character depending on the nature of the interacting species. In simple chemical terms, the stability of a system enjoying collective bonding cannot be justified in terms of 1,2 interactions alone. In extreme cases, an interesting regime in which the covalent terms change their usual short-range nature to longer-ranged one appears, and the covalent contributions start resembling the electrostatic ones in ionic crystals and need to be considered globally.

## Methods

All structures were optimized at the closed-shell and broken symmetry M06-2X[32]/def2TZVPP[33] computational levels. The M06-2X functional was selected since our previous study showed that it can reproduce accurate geometries for the $[M^{n+}BH_3]^{2-n}$ species. It is also one of the four DFT functionals implemented in the AIMAll[34] package for IQA analyses[16–19]. The nature of the local minima was examined by frequency analysis and only local minima with the $C_{3v}$ point groups were selected for further study. All first-principles computations were performed with the Gaussian 16. Rev. B01 suite of programs[35]. The Kohn-Sham wave functions obtained from the DFT computations were further analyzed within the framework of the quantum theory of atoms in molecules (QTAIM)[11] and the IQA energy decomposition approach[36,37]. Fuzzy, multiconfigurational and CC IQA calculations were performed with the PROMOLDEN code[38]. Fuzzy atoms were defined according to Becke's prescription[39] with Slater-Bragg radii and iteration level $k = 3$[40]. The CC IQA calculations were performed using the BBC1 density matrix functional approximation for the exchange-correlation density[41,42]. The computational data produced in this study are available in Supporting Information free of charge.

## Data availability

Dissociation, promotion, and deformation energies for $MAX_3$ complexes, the list of the contributions of interatomic exchange-correlation and electrostatics for the species presented in Fig. 2, and full IQA energy components computed at DFT and post-HF levels also IQA within fuzzy atom partitioning for selected compounds, plots of HOMO for selected molecules, and AdNDP plots for $NaBH_3^-$ are deposited in the Supporting Information free of charge. The cartesian coordinates of all species are deposited in Supplementary Data 1 in *.XLSX format.

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

## Acknowledgements

We thank Shant Shahbazian for detailed discussions during the preparation of the manuscript. MK acknowledges the financial support by the Natural Sciences and Engineering Research Council of Canada (NSERC) and Canada Research Chairs Program. Computational resources for the first-principles computations were provided by Compute Canada, SharcNet, and by the project "e-Infrastruktura CZ" (e-INFRA CZ LM2018140) supported by the Ministry of Education, Youth and Sports of the Czech Republic. AMP thanks the Spanish MICINN, grant PGC2018-095953-B-I00. CFN thanks to National Science Centre, Poland 2020/39/B/ST4/02022 for funding this research. For the purpose of Open Access, the author has applied a CC-BY public copyright license to any Author Accepted Manuscript (AAM) version arising from this submission.

## Author contributions

S.S.H., V.Š., S.A.S., and M.K. performed computations and analyzed data. M.A.P., and C.F.N. developed the idea and wrote the paper.

## Competing interests

The authors declare no competing interests.
