## [Peer Review File · Nature Communications]

REVIEWER COMMENTS

Reviewer #1 (Remarks to the Author):

The present work introduces the concepts of collective bonding and collective index. The work is interesting and imaginative. However, in my opinion, the conclusions achieved strongly depends on the method used to partition the space into the atomic components. I am sure the results will be different if the authors use ELF basins instead of AIM basins or Voronoi or Hirshfeld or a fuzzy partition of space. I am not confident at all that the conclusions derived from IQA using the AIM partition will be the same using other partitions of the space. So, maybe these collective bonding is an artifact of how the space is partitioned in AIM theory. On the other hand, as a chemist, I would like to understand these collective bonds. Which fragment orbitals are involved in the bonding? Is there any charge transfer? What are the mechanism of stabilization? New concepts are welcome but only when they are really needed. I do not think we need to add noise to the already noisy field of the chemical bond with concepts like collective bonding and collective index. I consider that the famous statement of Coulson in 1959 ("Give me insight, not numbers") is not followed in this work. For this reason, my recommendation is to not publish this work in Nat. Commun. The authors may also consider to answer the following points when they submit the work in another journal:

1. It seems to me that the correct formula for NaBH_3^- and related compounds is $\text{Mn}^+\text{BH}_3^{n-3}$ and not $\text{Mn}^+\text{BH}_3^{2-n}$, so for $n = 1$ like in the case $\text{M} = \text{Na}$, we have $\text{Na}^+\text{BH}_3^{-2} = \text{NaBH}_3^-$.
2. p. 5, when discussing Figure 1. The -94.4 kcal/mol threshold is marked by a dashed red line (and not blue).
3. p. 8. The fact that charge shift bonds as in F_2 have a higher $\text{VXC}(\text{A},\text{B})$ component is difficult to justify taking into account that charge shift bonds derive their stability from the resonance of ionic forms rather than the covalent sharing of electrons. Exchange-correlation contribution of resonant structure like F^+F^- should go to $\text{VXC}(\text{A},\text{A})$ and, consequently, $\text{VXC}(\text{A},\text{B})$ should be smaller in charge shift bonds than in classical covalent bonds.
4. Table 2. Let's consider the chemical bond in LiCH_3 . Should we consider that hydrogen atoms are bonded to C and at the same time to Li but Li is not bonded to C? If we accept this interpretation, then we should change all chemistry books. Maybe we will have to do it, but before, if we have to accept this change of paradigm, it should be put on more firmly grounds. Let me add that a previous work shows that the delocalization index with CH_3 and Li in LiCH_3 is basically the same as between C and Li (see E. Matito et al. J. Phys. Chem. B, 110 (2006) 7189-7198), meaning that the interaction of H and Li is minor. This result is completely different to the result presented in this paper and it is much more reasonable.

Reviewer #2 (Remarks to the Author):

The paper of Foroutan-Nejad et al. is a very interesting work which reopens many questions concerning the bonding situation in several molecules for which it is widely accepted that the bonding nature is completely known and understood. The authors introduced a novel family of chemical bonds and call it collective bonds. The presented results are based on the interacting quantum atoms (IQA) energy partitioning scheme. It is known that the energy partition can be performed using several different methods. In addition, there other more elaborate theoretical treatment (VB, for instance) which can provide an adequate insight into bonding nature. I am sure that the present paper will initiate a vivid discussion on this new bonding concept and on our general knowledge on chemical bonding. The presented paper is well-written, well-structured, and thus I would recommend acceptance of the manuscript. Anyway, I would suggest the following minor point to be amended:

1. The authors should quote two very related papers: Chemical Physics Letters 698 (2018) 19–23 (<https://doi.org/10.1016/j.cplett.2018.03.007>) and J. Phys. Chem. A 2020, 124, 26, 5369–5377 (<https://doi.org/10.1021/acs.jpca.0c03432>).

Reviewer #3 (Remarks to the Author):

In this work, Shahbazian, Foroutan-Nejad, and co-workers introduce a novel family of chemical bonds called collective bonds. The authors report that this new type of bond occurs in a large family of compounds, some of them used (or stored) in laboratories. The compounds analyzed have the general formula MCR_3 and aluminum-boron complexes with general formula MAH_3 (A: B or Al). After reading the paper with great interest and reviewing the background, I appreciate that the justification for introducing this new type of bond is not adequately elaborated.

Firstly, I would like to refer to the $NaBH_3$ - system, which, as the authors mention, the classification of its bond has been a source of much debate in the last two years. One of the current authors introduced the idea of cooperative bonds, i.e., the covalence between Na and B should be enabled by attractive electrostatic interactions between the H (BH_3) and Na. Therefore, from this perspective, one could say that this type of bonding was previously introduced in the literature (Angew. Chem. 2020, 132, 1 - 5), and the contribution of the current work would only be a generalization of this previous work.

Secondly, I missed a reflection about the weakness of theoretical methods for analyzing chemical bonding in complicated systems (i. e. NaBH_3^-). As outlined by Pino-Rios et al., "The physical nature of the bond is not a puzzle, but rather, it comes from the interference of the atoms' wave functions. What makes the NaBH_3^- bond unusual is the difficulty of describing it with standard bonding models" (Angew. Chem. Int. Ed. 2021, 60, 12747–12753). In the same sense, the first thing I would have liked in this work is a comparative analysis between the works that have converged in an approximate description of the chemical bonding in the NaBH_3^- system. However, this is not done; on the contrary, the authors make some errors in the general comparisons. For example, they say that spin-polarized bond in NaBH_3^- , proposed by Salvador et al. (Angew. Chem. Int. Ed. 2020, 59, 1 – 6), has a contribution of 29% in the work of Radenkivi et al., who analyzed the NaBH_3^- bond in the framework of Valence bond theory (Angew. Chem. 2021, 133, 12833 –12836), when in fact it only corresponds to 9.9% [3.6 kcal/mol (spin-polarized) vs. 36.4 kcal/mol (total interaction energy)].

Finally, at equilibrium, the NaBH_3^- system presents a high multireference character (Angew. Chem. 2020, 132, 2 – 7). Therefore, it is necessary to analyze its correlated wave function. The wave function used in this work (broken symmetry) is an approximation, taken from the reference of Salvador et al. However, its pertinence is not justified. Other related systems (MBH_3), studied here, will also present substantial multi-referential character. Based on this analysis, I consider that this work's current version does not meet the conditions to be accepted in Nature communications.

Response to Reviewers

Reviewer #1 (Remarks to the Author):

Reviewer: The present work introduces the concepts of collective bonding and collective index. The work is interesting and imaginative. However, in my opinion, the conclusions achieved strongly depends on the method used to partition the space into the atomic components. I am sure the results will be different if the authors use ELF basins instead of AIM basins or Voronoi or Hirshfeld or a fuzzy partition of space. I am not confident at all that the conclusions derived from IQA using the AIM partition will be the same using other partitions of the space. So, maybe these collective bonding is an artifact of how the space is partitioned in AIM theory. On the other hand, as a chemist, I would like to understand these collective bonds. Which fragment orbitals are involved in the bonding? Is there any charge transfer? What are the mechanism of stabilization? New concepts are welcome but only when they are really needed. I do not think we need to add noise to the already noisy field of the chemical bond with concepts like collective bonding and collective index. I consider that the famous statement of Coulson in 1959 ("Give me insight, not numbers") is not followed in this work. For this reason, my recommendation is to not publish this work in Nat. Commun. The authors may also consider to answer the following points when they submit the work in another journal:

Response:

The reviewer states that in his/her opinion, the conclusions strongly depend on the method. To check if that is the case, we repeated all computations using fuzzy atom-based IQA. The final conclusions remain the same and thus are not dependent on the method. There are systems in which interaction is not between formally bonded 1,2 atoms but instead 1,3 interactions define the bonds. Importantly, the MO picture cannot recover collective bonds and this is the main reason these bonds have remained unnoticed so far. Collective bonds are unlike "multicenter bonds" in which some MOs spread all over several atoms. In collective bonds MOs partially penetrate into atomic basins and cause collective bonds to form. Changing atomic partitioning methods marginally affects the magnitude of ICI, the collective bond indices, but the main picture remains the same. In a subsequent contribution we will study the effect of changing the MO picture on collective bonding, i.e. localization of MOs also affects the magnitude of ICI, but again the main picture does not change. However, this issue is beyond the present work.

Reviewer: 1. It seems to me that the correct formula for NaBH_3^- and related compounds is $\text{M}^{n+}\text{BH}_3^{2-n}$ and not $\text{M}^{n+}\text{BH}_3^{n-3}$, so for $n = 1$ like in the case $\text{M} = \text{Na}$, we have $\text{Na}^+\text{BH}_3^{2-1} = \text{Na}^+\text{BH}_3^-$.

Response:

This is true as it is presented for $\text{M}^{(n+)}\text{AlH}_3^{(2-n)}$ species that are introduced in the introduction. We used a simplified formula as MAIH_3 and MBH_3 throughout the manuscript.

Reviewer: 2. p. 5, when discussing Figure 1. The -94.4 kcal/mol threshold is marked by a dashed red line (and not blue).

Response: Corrected.

Reviewer: 3. p. 8. The fact that charge shift bonds as in F₂ have a higher V_{XC}(A,B) component is difficult to justify taking into account that charge shift bonds derive their stability from the resonance of ionic forms rather than the covalent sharing of electrons. Exchange-correlation contribution of resonant structure like F⁺F⁻ should go to V_{XC}(A,A) and, consequently, V_{XC}(A,B) should be smaller in charge shift bonds than in classical covalent bonds.

Response:

Without entering the charge-shift bonding (CSB) debate, it is of utmost importance to recognize that the resonance of “ionic” forms in F₂ is zwitterionic, i.e., the two resonating forms are equally probable unlike in the traditional ionic bonding in, e.g., LiF. When this is acknowledged, the larger V_{XC}(A,B) is easily understood. In an extreme case, for instance for the H⁽⁺⁾-H⁽⁻⁾

↔ H⁽⁻⁾-H⁽⁺⁾ resonance at large R(H,H), the interatomic delocalization index rises to 2, and with it the value of V_{XC}(A,B). The latter is, to first order, equal to $-\Delta(A,B)/(2RAB)$, where Δ is the A,B delocalization index. An intuitive way to understand this is to recall that V_{XC} is obtained by integrating the exchange-correlation density divided by the interelectron distance. When the electrons lie in two different regions, this distance can be taken to the first order as a constant, RAB, such that only the exchange-correlation density is actually integrated, giving rise to $\Delta(A,B)$. The latter is a measure of the interatomic fluctuation of the electron populations. It is quite intuitive that for, e.g., a 2c-2e link the fluctuation peaks when either of the two electrons is in one region or they are in the other with equal probability. This is exactly the case of a zwitterionic resonance. So, although counterintuitive at first, in a CSB the XC contribution does not concentrate on the V_{XC}(A,A) and V_{XC}(B,B) terms but, quite on the contrary, on the V_{XC}(A,B) one. This has been considered in depth in several papers, for example see

Martín Pendás, A.; Francisco, E. Decoding Real Space Bonding Descriptors in Valence Bond Language. *Phys. Chem. Chem. Phys.* **2018**, *20* (18), 12368–12372 DOI: 10.1039/c8cp01519h.

Casals-Sainz, J. L.; Jiménez-Grávalos, F.; Francisco, E.; Martín Pendás, A. Electron-Pair Bonding in Real Space. Is the Charge-Shift Family Supported? *Chem. Commun.* **2019**, *55* (35), 5071–5074 DOI: 10.1039/c9cc02123j.

Reviewer: 4. Table 2. Let's consider the chemical bond in LiCH₃. Should we consider that hydrogen atoms are bonded to C and at the same time to Li but Li is not bonded to C? If we accept this interpretation, then we should change all chemistry books. Maybe we will have to do it, but before, if we have to accept this change of paradigm, it should be put on more firmly grounds. Let me add that a previous work shows that the delocalization index with CH₃ and Li in LiCH₃ is basically the same as between C and Li (see E. Matito et al. *J. Phys. Chem. B*, *110* (2006) 7189-7198), meaning that the interaction of H and Li is minor. This result is completely different to the result presented in this paper and it is much more reasonable.

Response:

We are absolutely grateful to the reviewer for pointing out this issue. We repeated the analysis and realized that the values for MCH₃ and some relevant compounds are wrong. The source of error was automatization of data processing. This issue has been corrected and re-checked.

Reviewer #2 (Remarks to the Author):

Reviewer: The paper of Foroutan-Nejad et al. is a very interesting work which reopens many questions concerning the bonding situation in several molecules for which it is widely accepted that the bonding nature is completely known and understood. The authors introduced a novel family of chemical bonds and call it collective bonds. The presented results are based on the interacting quantum atoms (IQA) energy partitioning scheme. It is known that the energy partition can be performed using several different methods. In addition, there other more elaborate theoretical treatment (VB, for instance) which can provide an adequate insight into bonding nature. I am sure that the present paper will initiate a vivid discussion on this new bonding concept and on our general knowledge on chemical bonding. The presented paper is well-written, well-structured, and thus I would recommend acceptance of the manuscript. Anyway, I would suggest the following minor point to be amended:

Response: We thank the reviewer for his/her positive response and feedback.

Reviewer: 1. The authors should quote two very related papers: Chemical Physics Letters 698 (2018) 19–23 (<https://doi.org/10.1016/j.cplett.2018.03.007>) and J. Phys. Chem. A 2020, 124, 26, 5369–5377 (<https://doi.org/10.1021/acs.jpca.0c03432>).

Response: We agree and have cited the above references in the revised manuscript.

Reviewer #3 (Remarks to the Author):

Reviewer: In this work, Shahbazian, Foroutan-Nejad, and co-workers introduce a novel family of chemical bonds called collective bonds. The authors report that this new type of bond occurs in a large family of compounds, some of them used (or stored) in laboratories. The compounds analyzed have the general formula MCR_3 and aluminum-boron complexes with general formula MAH_3 (A: B or Al). After reading the paper with great interest and reviewing the background, I appreciate that the justification for introducing this new type of bond is not adequately elaborated.

Response:

We respect the reviewer's general opinion, but we clearly do not share it. As the considerable literature that has accumulated over the last years on the $NaBH_3^-$ anion has shown, the rationalization of the chemical bonding in this and, as we now show, similar compounds, is far from settled. As it happens with some other controversial bonding situations uncovered in recent times (e.g. the infamous quadruple bond in C_2), different flavors of chemical bonding analyses provide answers that can lead to contradictory opinions and long academic discussions. What we try to show here (convincingly in our opinion) is that using real space orbital invariant techniques, which are as free from theoretical level artifacts as possible, a rather clear pattern, which had already been pointed out in $NaBH_3^-$ but is now found to be much more general, appears. According to real space descriptors which are obtained at various theoretical levels and with several spatial partitioning, in some systems there is no clear justification to consider bonding as a set of separated "bonds", be them two- or multi-centered ones. It is important to distinguish the "collective" bonding introduced here from multi-centered bonds. In the latter, $V_{XC}(A,B)$ decays very quickly with the A-B distance, so that energetically speaking covalent contributions are still very short-ranged. In "collective" systems, far away energetic contributions are essential for the cohesion of the system. This is, in our opinion, new.

In the following, we provide answers to all of the reviewer's questions and concerns.

Reviewer: Firstly, I would like to refer to the $NaBH_3^-$ system, which, as the authors mention, the classification of its bond has been a source of much debate in the last two years. One of the current authors introduced the idea of cooperative bonds, i.e., the covalence between Na and B should be enabled by attractive electrostatic interactions between the H (BH_3) and Na. Therefore, from this perspective, one could say that this type of bonding was previously introduced in the literature (Angew. Chem. 2020, 132, 1 - 5), and the contribution of the current work would only be a generalization of this previous work.

Response:

The present work has two important messages. First, as the reviewer pointed out, this work has generalized the results obtained for a single molecule that has been detected in gas-phase to a vast group of molecules that are present in chemistry labs in bulk. We believe that it is important to publish this generalization in a journal with diverse readers because some of the molecules

that are suggested to form collective bonds are ordinary reagents in organic and inorganic labs. The second point that is entirely new and was not discussed in the previous work is that apart from certain molecules in which 1,2 interactions are repulsive while 1,3 interactions are attractive, we have a large group of molecules in which 1,2 interactions (whether repulsive or slightly attractive) are significantly less efficient than 1,3 interactions. This picture cannot be obtained from MO theory because no multicenter MO can be found between 1,3 interacting atoms. Nevertheless, the morphology of MOs depends on their selected isosurface values and localizations. Real-space partitioning however, verifies these unusual bonds. In this work we have shown that the outcome is robust and our analysis does not change by changing the partitioning method from QTAIM, which has atoms with solid boundaries, to fuzzy atom partitioning having atoms with undefined borders (IQA). We firmly believe that what we observe is a real effect and deserves to be publicized in a journal with a broad and diverse readership.

Reviewer: Secondly, I missed a reflection about the weakness of theoretical methods for analyzing chemical bonding in complicated systems (i. e. NaBH_3^-). As outlined by Pino-Rios et al., "The physical nature of the bond is not a puzzle, but rather, it comes from the interference of the atoms' wave functions. What makes the NaBH_3^- bond unusual is the difficulty of describing it with standard bonding models" (Angew. Chem. Int. Ed. 2021, 60, 12747–12753). In the same sense, the first thing I would have liked in this work is a comparative analysis between the works that have converged in an approximate description of the chemical bonding in the NaBH_3^- system. However, this is not done; on the contrary, the authors make some errors in the general comparisons. For example, they say that spin-polarized bond in NaBH_3^- , proposed by Salvador et al. (Angew. Chem. Int. Ed. 2020, 59, 1 – 6), has a contribution of 29% in the work of Radenkivi et al., who analyzed the NaBH_3^- bond in the framework of Valence bond theory (Angew. Chem. 2021, 133, 12833 –12836), when in fact it only corresponds to 9.9% [3.6 kcal/mol (spin-polarized) vs. 36.4 kcal/mol (total interaction energy)].

Response:

We thank the reviewer for pointing out this issue. We have fixed this point in the new revision.

Reviewer: Finally, at equilibrium, the NaBH_3^- system presents a high multireference character (Angew. Chem. 2020, 132, 2 – 7). Therefore, it is necessary to analyze its correlated wave function. The wave function used in this work (broken symmetry) is an approximation, taken from the reference of Salvador et al. However, its pertinence is not justified. Other related systems (MBH_3), studied here, will also present substantial multi-referential character. Based on this analysis, I consider that this work's current version does not meet the conditions to be accepted in Nature communications.

Response:

We performed T-test analysis at CCSD(T) level on all studied species. We found only 7 molecules with ill-defined wave functions. All these molecules were analysed at three different levels of theory; after optimization of their geometries at CCSD level we performed IQA analysis at the same level. Then DFT-IQA analysis at the CCSD geometries were performed and then CASSCF IQA on CASSCF-optimized geometries were performed to assess the effects of dynamic and static correlation on the outcome of the IQA analysis. All data are in good agreement with the original DFT-based analysis. Changing the computational level changes the quantitative values as one expects but the main characteristics like small ICI values that denote strong 1,3 interactions versus weak 1,2 interactions persists. We hope our analyses convince the respected referee that our data are legitimate and worth getting published in Nature Communications.

REVIEWER COMMENTS

Reviewer #1 (Remarks to the Author):

I appreciate the effort done by the authors to improve the paper. Still, there are some points that should be solved before recommending publication:

1) As defined in the paper, species with collective interactions have either weakly stabilizing or even completely destabilizing M-A bonds. Now, if one considers the pyramidal and the inverted structures, all these species should have an inverted structure which is either more stable than the pyramidal one or at least of similar energy than the pyramidal one. I think it is important to have for all species with collective interactions, the relative energies of the pyramidal and inverted structures. According to Table 1, only the $i\text{-CaAlH}_3$ is more stable than the pyramidal counterpart. In fact, there are many species with collective interactions that do not have an inverted structure. If the reason for the bonding is the formation of the M-X bonds, we should have inverted structures for all species with collective interactions. On the other hand, species that should not be considered collective bonded like LiCF_3 , NaCF_3 or $\text{CaC}(\text{CH}_3)_3$, they have inverted minima. This is an unexpected result.

2) The authors conclude that collective interactions originate from penetration of a substantial part of the HOMO into the atomic basin of the X atoms of the MAX_3 systems. If this is so, then the conclusions achieved should strongly depend on how atomic basins are defined and, therefore, on the method used to partition the space into the atomic components. Although they have analyzed the fuzzy partition of space finding similar results than the QTAIM partition (except for some case like ICIC of NaAlH_3^- and KBH_3^-), I am not confident at all that the conclusions derived from IQA using the AIM partition will be the same using other partitions of the space. So, maybe these collective bonding is an artifact of how the space is partitioned in AIM theory.

3) What is the dissociation energy of the MAX_3 systems? Dissociation energies should be provided.

4) On the other hand, as a chemist, I would like to understand these collective bonds. Which fragment orbitals are involved in the bonding? Is there any charge transfer? What is the mechanism of stabilization?

5) The AdNDP picture of the NaBH_3^- bonds is the classical 2c-2e bond picture. The authors write that the MO of the Na-B bond invades the H atomic basins. I do not see such invasion in the picture.

The authors may also consider to answer the following minor points:

1. It seems to me that the correct formula for NaBH_3^- and related compounds is $\text{M}^{\text{n}+}\text{BH}_3^{\text{n}-3}$ or $[\text{M}^{\text{n}+}\text{BH}_3]^{\text{n}-3}$ but not $\text{M}^{\text{n}+}\text{BH}_3^{\text{n}-2}$. With the formula $\text{M}^{\text{n}+}\text{BH}_3^{\text{n}-2}$ for $n = 1$ like in the case $\text{M} = \text{Na}$, we have $\text{Na}^+\text{BH}_3^- = \text{NaBH}_3$ and not NaBH_3^- .
2. p. 6, define deformation energy and provide values of these deformation energies for all systems.
3. p. 7. According to Table S1, the “-149.0” should be “133.9” and the “-21.0” should be “23.6”.
4. Tables 2, S3, and S4. Define T.
5. Typo: “valance”.

Reviewer #3 (Remarks to the Author):

The authors have revised their manuscript, and there is an effort to answer the criticisms or recommendations. After re-reading the paper, I feel that there is some disconnection between the interpretation of this work (based mainly on IQA analysis, which is the principal methodology of this paper) with the alternative interpretations. Especially for NaBH_3^- , for which, as commented by the authors, there are at least six papers in *Angewandte Chemie* with alternative interpretations. To make my point clear, NaBH_3^- , was initially interpreted as a dative $[\text{Na}:\text{BH}_3]^-$ bond (*Angew. Chem. Int. Ed.* 2019, 58, 13789; *Angew. Chem.* 2019, 131, 13927). However, it is later claimed to be a covalent $[\text{Na}-\text{BH}_3]^-$ bond in the equilibrium structure (*Angew.Chem.Int.Ed.*2020,59,8756-8759). Subsequently, the study of some of the authors of this work using IQA analysis predicts a covalent Na-B interaction promoted by electrostatic interactions between H (from BH_3) with Na (*Angew. Chem. Int. Ed.* 2020, 59, 20900- 20903). The two most recent papers, one based on valence bond theory (VBT), *Angew. Chem. Int. Ed.* 2021, 60, 12723-12726, and the other on electronic localization function(ELF) analysis, *Angew. Chem. Int. Ed.* 2021, 60, 12747, return to the interpretation of bonding using familiar concepts such as hybridization, orbital overlap, and electron pair distribution. TEV predicts a strong electrostatic interaction at equilibrium that binds Na to BH_3 , due to a dipole-dipole interaction between the dipole induced in Na by the polarization of one of its electrons (to the opposite side of B) and the permanent B-H dipole.

Nevertheless, TEV also finds an appreciable contribution due to the bonding character of the other electron, which is interpreted as a mixture between one-electron Na-B covalent bonding and a spin-polarized contribution, $\cdot\text{Na} \cdot\text{BH}_3$ (which is minimal), all these thanks to a combination of 3s-3p orbitals of Na at equilibrium. In the ELF case, the interpretation is that at equilibrium, the Na: goes to a 3s-3p hybridization (preparation energy) due to the BH_3 perturbation, the undistorted BH_3 (in contrast to the case of $\text{H}_3\text{N}:\text{BH}_3$, where BH_3 goes from sp^2 to sp^3 at equilibrium) would have the free p_z orbital available to receive Na- electrons. Then, the two Na- electrons are polarized between

the 3s-3p orbitals, one of them is located at one end (as a lone electron), and the other one is delocalized between the 3s-3p of Na and the pz of B, the authors call this a two-center one-electron bond (Na-B). Therefore, the electrostatic interaction is a consequence of the rearrangement of the electron density in the bonding situation. This rearrangement can be rationalized using chemist-friendly concepts and quantified by partitioning methods such as EDA, VBT, or IQA. The strong dipole-dipole interaction (or secondary electrostatics, according to IQA) would compensate for the energy required for the 3s-3p hybridization of Na that allows it to polarize its electrons. In this context, I consider that the interpretations related to the chemical bonding concept should always consider reviewing the classical models. As I understand it, what is seen with IQA is entirely compatible with what is interpreted by resorting to more traditional schemes, as long as methodologies that account for the particularities in the wave function of each system are used (the criticism of using EDA to NaBH₃⁻, was mainly supported by the non-correlated wave function used in this analysis).

In conclusion, I consider that this work shows an interesting analysis with good methodological support on the chemical bonding of the species: Ma+AR₃b⁻ (A: C, B or Al and M=Li-K and Be-Ca); however, the way it is presented is very technical, I consider that it will be difficult to understand by any chemist and even more disconnected from the more traditional ways of interpreting the chemical bond.

Other suggestions:

On page 4, lines 91-95: "Radenkovic et al.⁶, using breathing orbital valence bond (BOVB) analysis, verified that only -3.6 kcal.mol⁻¹, i.e., 9.9% ..." It should be clarified that it corresponds to the biradical dissociation mechanism (to Na· and [·BH₃]⁻), and -3.6 kcal.mol⁻¹ is the contribution called "spin-exchange covalent bonding mechanism" by Radenković et al. which corresponds to the "spin-polarized bond" claimed by Salvador et al. [\cdot N ·BH₃]⁻ as the unique bonding character of this system.

On page 9, line 199, change "valance" by valence

Response to the Reviewers

Response to Reviewer 1:

*Q1. As defined in the paper, species with collective interactions have either weakly stabilizing or even completely destabilizing M-A bonds. Now, if one considers the pyramidal and the inverted structures, all these species should have an inverted structure which is either more stable than the pyramidal one or at least of similar energy than the pyramidal one. I think it is important to have for all species with collective interactions, the relative energies of the pyramidal and inverted structures. According to Table 1, only the *i*-CaAlH₃ is more stable than the pyramidal counterpart. In fact, there are many species with collective interactions that do not have an inverted structure. If the reason for the bonding is the formation of the M-X bonds, we should have inverted structures for all species with collective interactions. On the other hand, species that should not be considered collective bonded like LiCF₃, NaCF₃ or CaC(CH₃)₃, they have inverted minima. This is an unexpected result.*

A1. As stated, molecular structure is the result of a subtle interplay among the energy gains and penalties that a system undergoes when their electrons take profit of either charge transfer, eventually leading to *ionic bonding*, or charge delocalization, grossly speaking leading to *covalent interactions*. The first is hampered by the ionization cost transforming AB into A⁺B⁻, and fostered by the $Q_A Q_B / R_{AB}$ electrostatic attraction left behind. The latter benefits from reduced fragments' kinetic energies induced by delocalization and is avoided, for instance, by large on-site electron repulsions. In this scenario, the IQA decomposition provides clear proxies for all these concepts in the deformation energy of the fragments and the electrostatic and exchange-correlation interaction terms. Interestingly, orbital thinking focuses on bonding/antibonding descriptors heavily based on overlap arguments, thus on delocalization, tending to by-pass the long-range character of electrostatic contributions, since usually these terms mostly cancel out due to electroneutrality (see, e.g. 10.1002/chem.201804160_for a longer discussion). This has led to a dichotomous language: we talk about bonds when covalency is relevant, and we add electrostatics only when this description encounters problems. For instance, we have no escape but dealing with an ionic LiF, but we keep on talking about a (polar) covalent O-H bond in water. It is in this sense that collective interactions are not well covered by the standard paradigm. Their chemical (covalent) bonds are collective. This doesn't mean that electrostatics does not play a role in determining whether inverted or pyramidal structures are found. It is simply saying, for instance, that on the pyramidal structures the expected M-A bonds should be basic contributors to bonding while they aren't.

Q2. The authors conclude that collective interactions originate from penetration of a substantial part of the HOMO into the atomic basin of the X atoms of the MAX₃ systems. If this is so, then the conclusions achieved should strongly depend on how atomic basins are defined and,

therefore, on the method used to partition the space into the atomic components. Although they have analyzed the fuzzy partition of space finding similar results than the QTAIM partition (except for some case like ICIC of NaAlH₃⁻ and KBH₃⁻), I am not confident at all that the conclusions derived from IQA using the AIM partition will be the same using other partitions of the space. So, maybe these collective bonding is an artifact of how the space is partitioned in AIM theory.

A2. We understand the reviewer's point, but we do not share it. We have used some orbital arguments in the manuscript to make it more readable and accessible to the general audience of the journal, *but our conclusions cannot and should not* be read in orbital terms. Orbitals are only useful in a one-electron picture, which may be, or not, close to reality. Insisting on casting an orbital invariant description, as that provided by any real space chemical bonding analysis, in orbital terms is an error. If the one-electron image is sensible, both descriptions will come to an agreement. Otherwise they do not need to. Notice, for instance, that the HOMO simply does not exist in an explicitly correlated wavefunction, let alone in a quantum Monte Carlo calculation. This said, we now consider the possible dependence of our results on the chosen partition. We should emphasize on a few points:

(a) Both the QTAIM and fuzzy atom approach suggest the same values for ICIXc of all species including KBH₃⁻ (0.431 vs. 0.411) and NaAlH₃⁻ (0.795 vs. 0.808). Please note that collective interaction is merely defined on the basis of the exchange-correlation energy component that is the sole measure of covalency. (b) The differences in the ICIC values does not question the validity of collective interaction but they show how sensitive the electrostatic energy component is to slight changes of atomic basin shape, unlike resilient ICIXc. Luckily, chemical bonding depends on covalency.

(c) In a totally nontechnical language we believe that we can agree with the reviewer that within the framework of HF/DFT if an MO is abnormally extended towards a nucleus called X, although the electrons formally do not belong to X, X has a non-negligible effect on the electron's kinetic and potential energy.

(d) MO-based analyses are obsolete once we turn to post-HF electron density, where HOMO is no longer even recognizable. Our post-HF analyses all confirm the same conclusion drawn from DFT. In other words, the electron density in species with collective bonding is extended towards the peripheral atoms. Similarly, when using fuzzy atoms that have no clear wall, unlike QTAIM atoms, we obtain the same values for ICIXc, therefore, the same picture of bonding. It is worth noting that in some cases, this extension of electron density is so remarkable that atomic interaction lines (as defined within the context of QTAIM) are seen in the molecular graph of the species. We used the HOMO-based explanation to make this issue more understandable for chemists that are educated in MO theory.

Q3. What is the dissociation energy of the MAX₃ systems? Dissociation energies should be provided.

A3. All values are now listed in the SI, Table S1.

Q4. On the other hand, as a chemist, I would like to understand these collective bonds. Which fragment orbitals are involved in the bonding? Is there any charge transfer? What is the mechanism of stabilization?

A4. Similarly to Q2, we first notice that if it were possible to offer a fully-fledged orbital explanation of collective bonding, this would have already been fully considered in the literature. It is not, but we have indeed made an effort to translate its novelty into conventional orbital thinking. Obviously, much is lost in translation. Offering a very brief summary of collective bonding to a chemist implies understanding that standard bonding is short-ranged (this short-sightedness is independent of the two- three- of n-center character of a standard pair bond). Much as chemists do not have problems in recognizing that electrostatic interactions are long-ranged, e.g. when they learn about the Madelung constant problem in fresh general chemistry courses, there are situations when *covalency* is also not short-ranged and the collective effect of the environment is needed to understand the stability of a system. On top of this, charge transfer may or may not actually exist. In the cases here presented it is also important, as evidenced by the electrostatic contributions. Now, if we cast forcefully this framework onto the orbital description. We already provided the MO justification of collective bonding, i.e., the extension of HOMO towards peripheral species. Collective bonding is not detectable by MO analysis. This is why unlike multi-center bonding, it has remained unrecognized by the chemical community. The mechanism of stabilization is very clearly explained within the context of IQA by dissecting the energy into meaningful components that are exchange-correlation and electrostatics. Components like charge-transfer, orbital interaction, and possibly Pauli repulsion that is widely used by chemical community are unfortunately often nonphysical and as we recently discussed “process functions” that can lead to wrong conclusions. IQA energy components are all state functions and directly related to the quantum mechanical energy components, i.e. exchange correlation and electrostatics. Please read two of our recent works on the problems of alternative energy decomposition approaches (PCCP, 2020, 22, 22459-22464 and PCCP, 2022, 24, 2344-2348).

Q5. The AdNDP picture of the NaBH₃⁻ bonds is the classical 2c-2e bond picture. The authors write that the MO of the Na-B bond invades the H atomic basins. I do not see such invasion in the picture.

A5. Again, we included the QTAIM based AdNDP picture of NaBH₃⁻ to offer a way to interpret our results in orbital language. It just helps rationalize the orbital results. Please see Figure S1 for representation of the atomic basins and the penetration of HOMO into it. AdNDP is not different. The reviewer should take into account that the real space H atoms of the BH₃ moiety are

negatively charged and occupy a considerable *volume*. A change of the spatial partition (Q2) that changes the H charge does also affect the AdNDP orbitals. However, as we have shown, this does not alter the collective index significantly!

Q6. It seems to me that the correct formula for NaBH_3^- and related compounds is $\text{Mn}^+\text{BH}_3^{n-3}$ or $[\text{Mn}^+\text{BH}_3]^{2-n}$ but not $\text{Mn}^+\text{BH}_3^{2-n}$. With the formula $\text{Mn}^+\text{BH}_3^{2-n}$ for $n = 1$ like in the case $\text{M} = \text{Na}$, we have $\text{Na}^+\text{BH}_3^- = \text{NaBH}_3$ and not NaBH_3^- .

A6. This is corrected.

Q7. p. 6, define deformation energy and provide values of these deformation energies for all systems.

A7. The deformation energy is defined exactly the same as any other theory. It is the energy difference between the energy of the fragments in the molecule with their relaxed nonbonded form. We added definition of deformation and “promotion” energies along with their values with respect to different reference states.

Q8. p. 7. According to Table S1, the “-149.0” should be “133.9” and the “-21.0” should be “23.6”.

A8. These are corrected now.

Q9. Tables 2, S3, and S4. Define T.

A9. T represents the rest of atoms. This is fixed.

Q10. Typo: “valance”.

A10. Fixed.

Response to Reviewer 3:

Q1. *The authors have revised their manuscript, and there is an effort to answer the criticisms or recommendations. After re-reading the paper, I feel that there is some disconnection between the interpretation of this work (based mainly on IQA analysis, which is the principal methodology of this paper) with the alternative interpretations. Especially for NaBH_3^- , for which, as commented by the authors, there are at least six papers in *Angewandte Chemie* with alternative interpretations. To make my point clear, NaBH_3^- , was initially interpreted as a dative*

[Na:BH₃]- bond (Angew. Chem. Int. Ed. 2019, 58, 13789; Angew. Chem. 2019, 131, 13927). However, it is later claimed to be a covalent [Na-BH₃]- bond in the equilibrium structure (Angew.Chem.Int.Ed.2020,59,8756-8759). Subsequently, the study of some of the authors of this work using IQA analysis predicts a covalent Na-B interaction promoted by electrostatic interactions between H (from BH₃) with Na (Angew. Chem. Int. Ed. 2020, 59, 20900- 20903). The two most recent papers, one based on valence bond theory (VBT), Angew. Chem. Int. Ed. 2021, 60, 12723-12726, and the other on electronic localization function(ELF) analysis, Angew. Chem. Int. Ed. 2021, 60, 12747, return to the interpretation of bonding using familiar concepts such as hybridization, orbital overlap, and electron pair distribution. TEV predicts a strong electrostatic interaction at equilibrium that binds Na to BH₃, due to a dipole-dipole interaction between the dipole induced in Na by the polarization of one of its electrons (to the opposite side of B) and the permanent B-H dipole.

Nevertheless, TEV also finds an appreciable contribution due to the bonding character of the other electron, which is interpreted as a mixture between one-electron Na-B covalent bonding and a spin-polarized contribution, ·Na ·BH₃ (which is minimal), all these thanks to a combination of 3s-3p orbitals of Na at equilibrium. In the ELF case, the interpretation is that at equilibrium, the Na: goes to a 3s-3p hybridization (preparation energy) due to the BH₃ perturbation, the undistorted BH₃ (in contrast to the case of H₃N:BH₃, where BH₃ goes from sp² to sp³ at equilibrium) would have the free p_z orbital available to receive Na- electrons. Then, the two Na- electrons are polarized between the 3s-3p orbitals, one of them is located at one end (as a lone electron), and the other one is delocalized between the 3s-3p of Na and the p_z of B, the authors call this a two-center one-electron bond (Na·B). Therefore, the electrostatic interaction is a consequence of the rearrangement of the electron density in the bonding situation. This rearrangement can be rationalized using chemist-friendly concepts and quantified by partitioning methods such as EDA, VBT, or IQA. The strong dipole-dipole interaction (or secondary electrostatics, according to IQA) would compensate for the energy required for the 3s-3p hybridization of Na that allows it to polarize its electrons. In this context, I consider that the interpretations related to the chemical bonding concept should always consider reviewing the classical models. As I understand it, what is seen with IQA is entirely compatible with what is interpreted by resorting to more traditional schemes, as long as methodologies that account for the particularities in the wave function of each system are used (the criticism of using EDA to NaBH₃-, was mainly supported by the non-correlated wave function used in this analysis).

In conclusion, I consider that this work shows an interesting analysis with good methodological support on the chemical bonding of the species: Ma⁺AR₃b⁻ (A: C, B or Al and M=Li-K and Be-Ca); however, the way it is presented is very technical, I consider that it will be difficult to understand by any chemist and even more disconnected from the more traditional ways of interpreting the chemical bond.

A1. We are thankful for the detailed comment by the reviewer. As they mentioned, the results of the IQA and VB are clearly compatible. We also agree that this work is pushing the boundary of our understanding away from classical interpretation of chemical bonding. Furthermore, as the reviewer has very nicely summarized, quite a number of authors have contributed over the last years to enlarge our knowledge about chemical bonding in these systems. And as they show, every method/technology used uncovers a different bonding aspect. We do not deny the important insights provided by each method, but stress instead that most analyses are blind to specific aspects of bonding. Just briefly, orbital arguments mostly skip the role of electron correlation, that is only re-introduced after painfully taking the method to the limit. VBT, on which the reviewer focuses, allows for very chemically appealing images, but the rationalizations it offers cannot escape Pauling's resonance picture, and it will pick up a mixture of structures (the one-electron covalent plus the spin-polarized one in the case of NaBH_3^-) that heavily depend on the atomic or fragment orbitals used. In the case of alkali atoms, the large spatial extension of the valence ns orbitals is well-known to lead to specific effects in VB treatments. For instance, since the $2s$ Li orbital invades (overlaps) considerable the H atom in LiH, this system is found as basically covalent in non-orthogonal VBT. However, the analysis of any real space scalar field, like the Laplacian of the electron density, shows that the electron distribution is quasi-spherical around Li, and that this system is much better understood in terms of an ionic Li^+ core interacting with a polarized H^- anion (see, e.g. [10.1039/B604983D](https://doi.org/10.1039/B604983D) for a discussion). Finally, insights from Ziegler-Rauk EDA are extremely dependent on how the fragments are chosen as well as on how well the system and the fragments are described at the single-reference level, among other possible criticisms, see below. The only difference between our approach and the conventional methods is that here we define the energy components irrespective of orbitals within the framework of atoms. QTAIM has been known and used since 1970s, and is being used more and more. In that sense we do not agree that this approach is not understandable by all chemists. QTAIM has not been in the textbooks of chemistry but in the past decade it is introduced in some books like *Advanced Organic Chemistry* by F. A. Carey since 2007 (fifth edition) that is why more organic chemists are using this approach. Furthermore, we are sure that the reviewer knows that MO-based analyses are valid only within the context of single-reference methods, i.e., HF and DFT. If we focus on post-HF approaches, AOs and MOs no longer are the same familiar entities that can be used for rationalizing the bonding unless we get information from post-HF and artificially reinterpret them within the context of HF/DFT. Density-based approach is however free from this issue.

Another difference between our approach and the one used by Radenkovic et al. is that VB cannot define the atomic origin of the electrostatic attraction between the metal and the BH_3 fragment because VB does not break the energy into atomic terms. We can identify the atomic/group effect on the energy of a system that is consistent with the spirit of modern chemistry in which reactivities are discussed in terms of functional groups.

Furthermore, the ELF analysis on NaBH_3 does not provide orbital-based information. The orbital based analyses are added beside the ELF to the work because ELF is blind to one-electron

bond, unless someone interprets the data to make it compatible with the already known information. One can do the same by QTAIM delocalization index and interpret it in favor of one-electron bonding. Nevertheless, we now know that one-electron bond is not enough for the formation of the complex based on both VB and IQA detailed analyses.

Performing analyses like IQA is straightforward using Gaussian, that is the most used software by chemical community and AIMAll, a fully automated user-friendly software. Therefore, we believe this approach will find its followers and will soon become a common tool in chemistry and chemical bonding.

Furthermore, we would like to bring the attention of the reviewer to a more important problem with “conventional” bonding analysis approaches such as EDA. Recently some of us showed that EDA approach is suffering from a serious problem, i.e., its energy components are process function, not state function. This issue is beyond the theoretical level that EDA is performed at (PCCP, 2020, 22, 22459-22464 and PCCP, 2022, 24, 2344-2348).

Q2. On page 4, lines 91-95: "Radenkovic et al.⁶, using breathing orbital valence bond (BOVB) analysis, verified that only $-3.6 \text{ kcal.mol}^{-1}$, i.e., 9.9% ..." It should be clarified that it corresponds to the biradical dissociation mechanism (to $\text{Na}\cdot$ and $[\cdot\text{BH}_3]^-$), and $-3.6 \text{ kcal.mol}^{-1}$ is the contribution called "spin-exchange covalent bonding mechanism" by Radenkovic' et al. which corresponds to the "spin-polarized bond" claimed by Salvador et al. $[\cdot\text{N} \cdot\text{BH}_3]^-$ as the unique bonding character of this system.

A2. The explanation is added.

Q3. On page 9, line 199, change "valance" by valence.

A3. This is fixed.

REVIEWERS' COMMENTS

Reviewer #1 (Remarks to the Author):

The authors have positively answered most of the remarks of my reports. A few, minor points, still need to be addressed:

- 1) A comment on the reason why, in species with collective bonding, the inverted structures are not always the most stable despite the fact that the M-A bonds are destabilizing (or weakly stabilizing) should be included in the text.
- 2) I guess charges in Table 1 are QTAIM charges. If so, this should be indicated.
- 2) Typo: mold.